# TGFα controls checkpoints in CNS resident and infiltrating immune cells to promote resolution of inflammation

Lena Lößlein [1,2], Mathias Linnerbauer [1], Finnja Zuber[1], Thanos Tsaktanis [1], Oliver Vandrey[1], Anne Peter[1], Franziska Panier[1], Julia Zissler[1], Vivienne Riekher[1], Tobias Bäuerle[3,4], Jannis Hanspach [3], Frederik B. Laun[3], Lisa Nagel[3], Lisa Mészáros[5], Friederike Zunke [5], Jürgen Winkler [5], Ulrike J. Naumann[1], Nora Schwingen[1], Emely Neumaier[1], Arthur Liesz [6,7], Francisco Quintana [8,9] & Veit Rothhammer [1] ✉

After acute lesions in the central nervous system (CNS), the interaction of microglia, astrocytes, and infiltrating immune cells decides over their resolution or chronification. However, this CNS-intrinsic cross-talk is poorly characterized. Analyzing cerebrospinal fluid (CSF) samples of Multiple Sclerosis (MS) patients as well as CNS samples of female mice with experimental autoimmune encephalomyelitis (EAE), the animal model of MS, we identify microglia-derived TGFα as key factor driving recovery. Through mechanistic in vitro studies, in vivo treatment paradigms, scRNA sequencing, CRISPR-Cas9 genetic perturbation models and MRI in the EAE model, we show that together with other glial and non-glial cells, microglia secrete TGFα in a highly regulated temporospatial manner in EAE. Here, TGFα contributes to recovery by decreasing infiltrating T cells, pro-inflammatory myeloid cells, oligodendrocyte loss, demyelination, axonal damage and neuron loss even at late disease stages. In a therapeutic approach in EAE, blood-brain barrier penetrating intranasal application of TGFα attenuates pro-inflammatory signaling in astrocytes and CNS infiltrating immune cells while promoting neuronal survival and lesion resolution. Together, microglia-derived TGFα is an important mediator of glial-immune crosstalk, highlighting its therapeutic potential in resolving acute CNS inflammation.

The central nervous system (CNS) exhibits unique immunological characteristics, but at the same time it is particularly vulnerable to tissue-destructive events such as autoimmune inflammation due to its limited regenerative capacity[1–4]. Indeed, CNS-intrinsic inflammation during both acute and progressive stages of Multiple Sclerosis (MS) has yet been incompletely defined and poses major challenges in the treatment of MS patients[5–8]. Regionally restricted CNS inflammation is established early in disease and driven by the interaction of glial and CNS-infiltrating cells which occurs, especially in progressive stages, behind a tightened blood-brain barrier (BBB), which renders therapeutic strategies ineffective due to their limited BBB penetration capacities[9,10]. The CNS inflammatory microenvironment is characterized by various tissue-degenerative as well as protective factors provided by microglia, astrocytes, and CNS infiltrating immune cells. Deciphering their interplay is crucial to understand the underlying pathology with the goal to address an important gap in current

therapeutic approaches developing novel treatment strategies for acute and progressive neuroinflammation[11,12].

In this context, we have previously identified the transforming growth factor alpha (TGFα) as a mediator of microglia to astrocyte crosstalk, which mediates protective functions and is dysregulated in MS patients[13,14]. TGFα is a polypeptide with mitogenic properties and belongs to the epidermal growth factor (EGF) family. It activates various signaling pathways through the EGF receptor (EGFR), initiating cascades that promote cell survival and proliferation[15–18]. However, its specific effects and target populations remain to be defined to explore its potential for therapeutic intervention in autoimmune CNS inflammation. In this light, we here identify the spatiotemporal regulation of TGFα production and fine-dissect its effector populations in autoimmune CNS inflammation, including astrocytes, microglia, oligodendrocytes, and CNS-infiltrating immune cells. Showing its decrease in the cerebrospinal fluid in MS patients, we demonstrate in a translational therapeutic approach that its intranasal application leads to penetration into the CNS and induces lesion resolution in acute and progressive neuroinflammation in a preclinical animal model of MS. Together, we here describe the cellular and spatiotemporal regulation of CNS-intrinsic TGFα and evaluate its potential to improve outcomes in autoimmune inflammatory CNS disorders.

## Results

### Spatio-temporal regulation of TGFα in autoimmune CNS inflammation

We have shown in a previous transcriptome analysis of astrocytes at the peak of experimental autoimmune encephalomyelitis (EAE), an animal model of MS, that microglia-derived TGFα is a key regulator of astrocyte polarization[13]. By inducing expression of TGFα in microglia, among others, aryl hydrocarbon receptor (AHR) signaling restricts pro-inflammatory microglial transcriptional responses during EAE[13]. Despite these findings, the spatio-temporal regulation of TGFα and its impact on cellular targets other than astrocytes have yet to be defined. To address this gap in understanding and to redefine our insight into the underlying mechanisms, we here induced EAE in *C57BL/6J* (wild type) mice and determined TGFα-expressing cell populations over the course of EAE. Microglia were the predominant producers of TGFα, not only in the brain and spinal cord, but also in the optic nerve of EAE mice, while astrocytes produced lesser amounts of TGFα (Fig. 1a–d, Suppl. Fig. 1a–g). To further validate this observation, we analysed both total and proportional TGFα production to characterize its cell-specific dynamics. We observed that microglial TGFα reached high levels during peak and recovery stages of EAE, with microglia representing the predominant source of TGFα in the murine CNS (Fig. 1e–h, Suppl. Fig. 1e). In addition to microglia, astrocytes, neurons and a small fraction of cells histologically not explicitly stained for also contributed to TGFα production, albeit to a lesser extent (Fig. 1h). During late stages, the overall TGFα production returned to baseline, with microglia showing the most pronounced decline, while astrocytes and neurons produced lesser amounts of TGFα with specific kinetics predominantly in early and very late disease stages (Fig. 1e–h, Suppl. Fig. 1d–f). Overall, TGFα was upregulated by microglia, and to a lesser extent by astrocytes and neurons during autoimmune inflammation. To further identify the fraction of histologically undefined cells producing TGFα, we analysed a publicly available single-cell RNA-sequencing dataset[19] of EAE mice at the peak of disease. This analysis revealed that microglia exhibited the highest levels of *Tgfa* expression, several other cell types, including monocytes, neutrophils, oligodendrocytes, astrocytes, endothelial cells, T cells, dendritic cells and neurons showed detectable, though lower, levels of *Tgfa* expression, highlighting the specific cellular sources of TGFα during autoimmune inflammation (Fig. 1i). This was supported by in vitro studies showing that pro-inflammatory stimuli increased *Tgfa* expression in primary

murine and human microglia as well as in astrocytes, while showing minimal expression in neurons and leading to decreased expression in oligodendrocytes (Suppl. Fig. 1h–l). Moreover, when examining microglia in the brain and spinal cord separately, we observed a uniform distribution with no relevant differences in subsets or heterogeneity between these regions (Suppl. Fig. 1m). Interestingly, the expression of TGFα in microglia partially overlapped with high expression levels of Ki67 (Fig. 1j), suggesting an association of a proliferative microglial phenotype with tissue-protective properties described in previous studies[20].

Next, to define potential responder populations of TGFα, we investigated the expression of the epidermal growth factor receptor (EGFR/ ErbB1), the receptor for TGFα, on immune cell subsets during EAE. Flow cytometric analyses revealed EGFR expression on multiple cell populations in the CNS in addition to astrocytes, including microglia, myeloid cells, oligodendrocytes (PDGFRα[+]/ O4[+]), dendritic cells, CD4[+] T cells, monocytes, and macrophages, while most of those cell types displayed specific expression kinetics over the course of EAE (Fig. 1k).

Together, these analyses reveal that microglia are the main source of TGFα in the CNS during autoimmune inflammation alongside astrocytes and neurons, while in addition to astrocytes, both CNS resident and infiltrating immune cells represent its potential responder populations.

### TGFα promotes protective effects during CNS inflammation

TGFα production in microglia has previously been shown to be relevant for lesion resolution using shRNA-mediated transient knock-down approaches[13]. To strengthen these findings, we used a CRISPR-Cas9-based genetic perturbation model to persistently knock-out TGFα expression in microglia (Fig. 2a). A non-coding lentivirus under the control of the *Itgam* (*CD11b*) promoter was used as control. This approach led to a knock-out of microglial TGFα production confirmed by qPCR, flow cytometry, and immunohistochemistry (Suppl. Fig. 2a–d). In line with previous observations using shRNA-mediated knock-down[13], microglial TGFα ablation (*CD11b::TGFα*) impaired recovery from EAE (Fig. 2b, Suppl. Fig. 2e, f). High-parameter flow cytometric analysis followed by dimensionality reduction revealed an increase in CNS infiltrating immune cells in microglia-specific TGFα deficient mice (*CD11b::TGFα*) (Fig. 2c) consisting of CD4[+] T cells, macrophages, pro-inflammatory monocytes, pro-inflammatory B cells as well as neutrophils during recovery stages (Fig. 2d, e). CNS infiltrating T cells subsets in *CD11b::TGFα* mice exhibited an increased pro-inflammatory phenotype compared to controls, characterized by elevated IL-17 and IFNγ production (Fig. 2f).

Additionally, TGFα inactivation caused alterations in CNS resident cells, including loss of oligodendrocyte lineage cells (OLC, O4[+]) and Olig2[+] oligodendrocytes (Fig. 2g, h, Suppl. Fig. 2g). Besides, we noted neuronal toxicity, characterized by an increase in SMI32 expression in the spinal cord, a marker for degenerating axons[21], (Fig. 2i, Suppl. Fig. 2g), as well as a reduction in myelinated white matter area (Fig. 2j, k), overall suggesting increased demyelination and axonal damage upon reduced microglia-derived TGFα. In addition to these changes, the diminishment of microglial TGFα was associated with a reduction in the number of NeuN[+] neurons, suggesting its role in neuronal survival (Fig. 2l, Suppl. Fig. 2g).

Moreover, when the disease was monitored until very late stages, *CD11b::TGFα* mice exhibited a sustained failure to recover (Suppl. Fig. 2h, i). While the numbers of neutrophils and macrophages were comparable between groups at this late timepoint, *CD11b::TGFα* mice exhibited an increase in pro-inflammatory monocytes, CD4[+] T cells, and B cells (Suppl. Fig. 2j, k). We also found an expansion of effector T cells and elevated iNOS production in monocytes, indicating sustained inflammatory and neurodegenerative activity comparable to changes observed during early recovery (Suppl. Fig. 2k, l).

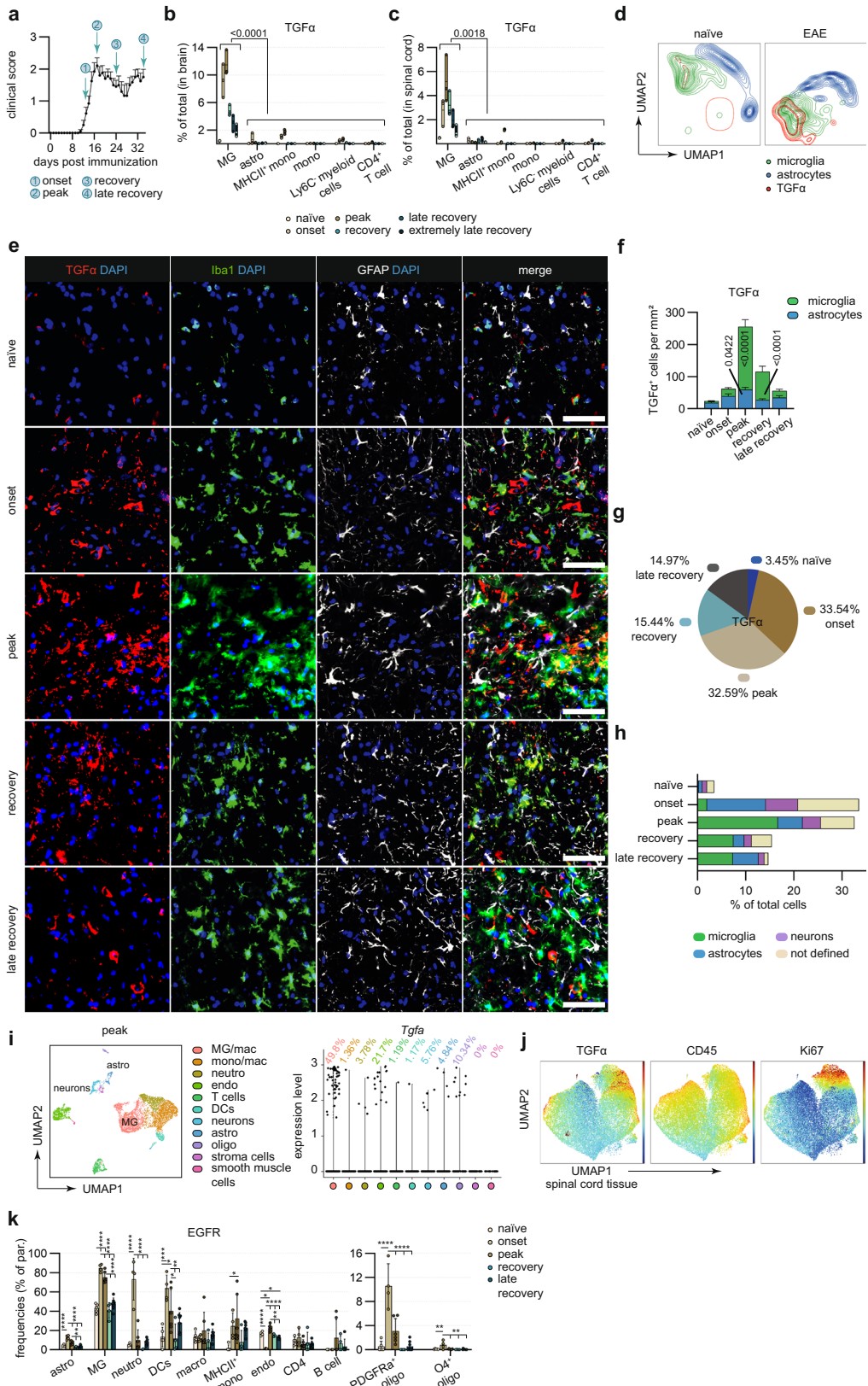

Collectively, these findings highlight the role of microglial TGFα for recovery from inflammatory damage in EAE due to its regulation of infiltrating T cells and pro-inflammatory myeloid cells in the CNS, as well as mitigation of OLC loss, demyelination, axonal damage, and neuronal loss.

## TGFα promotes protective effects under inflammatory and demyelinating conditions

These in vivo studies suggested pleiotropic effects of TGFα during autoimmune CNS inflammation. We thus examined distinct populations of TGFα responder cell types in the CNS in detail. To this end, we

**Fig. 1 | Spatio-temporal regulation of microglia-derived TGFα during acute CNS inflammation. a** EAE development and timepoints (onset, peak, recovery, late recovery) used for immunohistochemical and flow cytometric analysis of CNS cells ($n = 3/5$ per group). **b** TGFα production (% of total) of microglia (MG), astrocytes (astro), monocytes (mono), pro-inflammatory (MHCII + ) monocytes and Ly6C-myeloid cells in brains and spinal cords (**c**) of EAE during onset, peak, recovery, late recovery, extremely late recovery and naïve mice ($n = 3$ per timepoint) quantified by intracellular flow cytometry. **d** UMAP plot of TGFα expression in subsampled CNS resident cells during course of EAE ($n = 5–6$ per timepoint) analysed by high-dimensional flow cytometry. **e** Immunostaining and quantification (**f**) of TGFα+ Iba1+ and TGFα + GFAP+ cells and DAPI for nuclear staining in spinal cords of EAE (onset, peak, recovery, late recovery; $n = 3/5$ per timepoint) and naïve mice ($n = 5$). Scale bar 50 μm. **g** Total TGFα production in spinal cords of EAE at onset, peak, recovery, late recovery, and naïve mice ($n = 5$ per timepoint) quantified by immunostaining. **h** Proportional TGFα+ microglia (Iba1 + ), astrocytes (GFAP + ), neurons (NeuN + ) (% of all cells, based on DAPI counts) in spinal cords of EAE at onset, peak, recovery, late recovery, and naïve mice ($n = 5$ per timepoint) quantified by immunostaining. **i** UMAP plot (left) of scRNA-seq dataset (Wheeler et al., 2020), showing cell clustering from CNS samples at peak of EAE including expression levels (right) of TGFα across distinct cell clusters: microglia/macrophages (MG/mac), monocytes/macrophages (mono/mac), neutrophils, endothelial cells, T cells, dendritic cells (DCs), neurons, astrocytes, oligodendrocytes, stromal cells, and smooth muscle cells. **j** UMAP plot of subsampled microglia in the spinal cord during course of EAE ($n = 5-6$ per timepoint) analysed by high-dimensional flow cytometry showing the expression levels of TGFα, CD45 and Ki67. **k** Relative EGFR expression (% of parent) of astrocytes (astro), microglia (MG), neutrophils (neutro), dendritic cells (DCs), macrophages (macro), pro-inflammatory (MHCII + ) monocytes, endothelial cells (endo), CD4 + T cells, B cells, PDGFRα+ and O4+ oligodendrocytes of EAE during onset ($n = 4$), peak ($n = 6$), recovery ($n = 4$), late recovery ($n = 8$) and naïve mice ($n = 5$) quantified by intracellular flow cytometry. *$P < 0.05$; **$P < 0.01$; ***$P < 0.001$; ****$P < 0.0001$. Data shown as mean ± SD. Data shown as mean ± SEM in (**a**, **f**). Data shown as mean (centre line) and mean ± SEM (**b**, **c**). One-way ANOVA with Dunnett's multiple comparisons test in (**b–l**).

first determined autocrine effects of TGFα on microglia. Pro-inflammatory activation of primary murine microglia together with recombinant TGFα resulted in a reduction of pro-inflammatory genes and EGFR abundance (Fig. 3a–c, Suppl. Fig. 3a). Analogous experiments in primary mouse astrocytes revealed that TGFα co-administration increased astrocytic *Egfr* and tissue-protective *Lif* expression, and reduced production of CCL2, NOS2, and CSF2 at mRNA and protein levels (Fig. 3d–f, Suppl. Fig. 3a). Inflammatory activation of primary mouse microglia or astrocytes prior to TGFα exposure did not abolish the protective effects of TGFα (Suppl. Fig. 3b, c).

Expression of EGFR on macrophages and dendritic cells implied their responsiveness to TGFα during acute CNS inflammation. Indeed, TGFα dampened the expression of co-stimulatory molecules such as CD80 and CD86 on bone marrow-derived macrophages (BMDMs) and dendritic cells (BMDCs). Additionally, TGFα decreased CD74 expression, a molecule associated with regulatory immune responses related to antigen presentation (Fig. 3g–i). In the presence of both TGFα and LPS, BMDMs also demonstrated up-regulation of EGFR, suggesting that the activation of EGFR might be involved in modulating the macrophage response under inflammatory conditions (Fig. 3h).

We next aimed to evaluate the effects of TGFα on oligodendrocyte lineage cells (OLC) differentiation and survival. Indeed, TGFα reduced gene expression associated with apoptosis and pro-inflammatory chemokine production in response to activation (Fig. 3j, k, Suppl. Fig. 3a). Furthermore, TGFα tend to maintain oligodendrocyte precursor cells (OPCs) in an early maturation stage and enhanced their survival during oligodendrocyte differentiation (Suppl. Fig. 3d–f). Finally, exposure of ex vivo cultures of retinae and optic nerves to TGFα reduced the loss of Olig2⁺ oligodendrocyte precursors in an inflammatory environment (Fig. 3l, m).

To further study the effects of TGFα on oligodendrocytes in vivo, we used a toxicity-induced demyelination model independent of autoimmune inflammation. In the presence of demyelinating stress induced by phosphoglyceride lysolecithin (LCP), TGFα protected optic nerve tissue from myelin loss (Fig. 3l, n). Injection of TGFα together with LCP decreased demyelination in the corpus callosum of C57Bl/6 J mice, supporting the beneficial effects of TGFα on OLC survival and myelination (Fig. 3o, p).

Finally, neuronal cell death induced by TNFα was reduced by pre-incubation with TGFα (Fig. 3q, r). This finding was supported in free-floating cultures labelled for RNA-binding protein with multiple splicing (RBPMS⁺) retinal ganglion cells in ex vivo cultured retinae subjected to IFNγ challenge concomitant to TGFα (Fig. 3l, s, t). Notably, TGFα preserved retinal neuron survival in both central and peripheral retinal areas in the presence of IFNγ (Fig. 3t).

Collectively, TGFα dampens the pro-inflammatory phenotype of myeloid cells and promotes the survival of neurons and oligodendrocytes in vivo. Acting on astrocytes, dendritic cells, macrophages, OLCs, and neurons, TGFα modulates their response to inflammatory or demyelinating stress, suggesting TGFα-mediated protective effects in CNS inflammation due to its multi-targeted actions.

## Therapeutic TGFα administration in CNS inflammation

In order to reduce relapses and disease progression, RRMS patients are provided with disease-modifying therapies (DMTs)[22,23]. The majority of these treatments target priming and activation of pro-inflammatory immune cells in the peripheral immune compartment, while strategies promoting lesion resolution and inhibiting inflammatory neurodegeneration within the CNS are limited[11,12]. Indeed, the inability of macromolecules and proteins to cross the inflamed blood-brain barrier (BBB) represents a major impediment to CNS-targeting therapeutic approaches[24]. We thus applied the non-invasive strategy of intranasal delivery to evaluate the therapeutic value and tissue-protective functions of TGFα to improve lesion resolution during CNS inflammation. Notably, this therapeutic approach has the ability to deliver small proteins or molecules into the CNS[25,26]. Indeed, intranasal delivery of TGFα led to phosphorylation and activation of the EGFR even in naïve mice without inflammatory BBB disruption, suggesting its penetration into the CNS upon intranasal application (Suppl. Fig. 4a).

To evaluate therapeutic effects of TGFα in vivo, we induced EAE in *C57Bl/6 J* mice, intranasally administered TGFα daily before symptom onset at a dosage similar to the one used in previous studies administering comparably-sized proteins[25,26], and assessed its clinical and immunological effects (Fig. 4a). Pre-symptomatic TGFα treatment reduced disease severity and promoted recovery from CNS inflammation (Fig. 4b), and diminished Gadolinium-enhancing inflammatory lesion volumes in the spinal cord as determined by magnetic resonance imaging (Fig. 4c). Intranasal TGFα administration did not result in measurably elevated TGFα levels in the brain. The short half-life of TGFα, comparable to that of TGFα-PE38 in plasma (~10–20 min)[27], along with the dilution effect within the CNS and the detection limits of the applied methodology, likely contributed to these findings. (Suppl. Fig. 4c), high-parameter flow cytometric analyses of the CNS revealed increased EGFR expression in astrocytes (Suppl. Fig. 4b), along with a reduced proliferative and pro-inflammatory profile of CD4⁺ T cells in the CNS of TGFα-treated mice (Fig. 4d). This was accompanied by reduced microglial reactivity concomitant with decreased production of GM-CSF by microglia in TGFα-treated mice (Fig. 4e, f). Of note, intranasal TGFα application did not affect the number of CD4+ and CD8 + T cells, dendritic cells, monocytes, or pro-inflammatory and

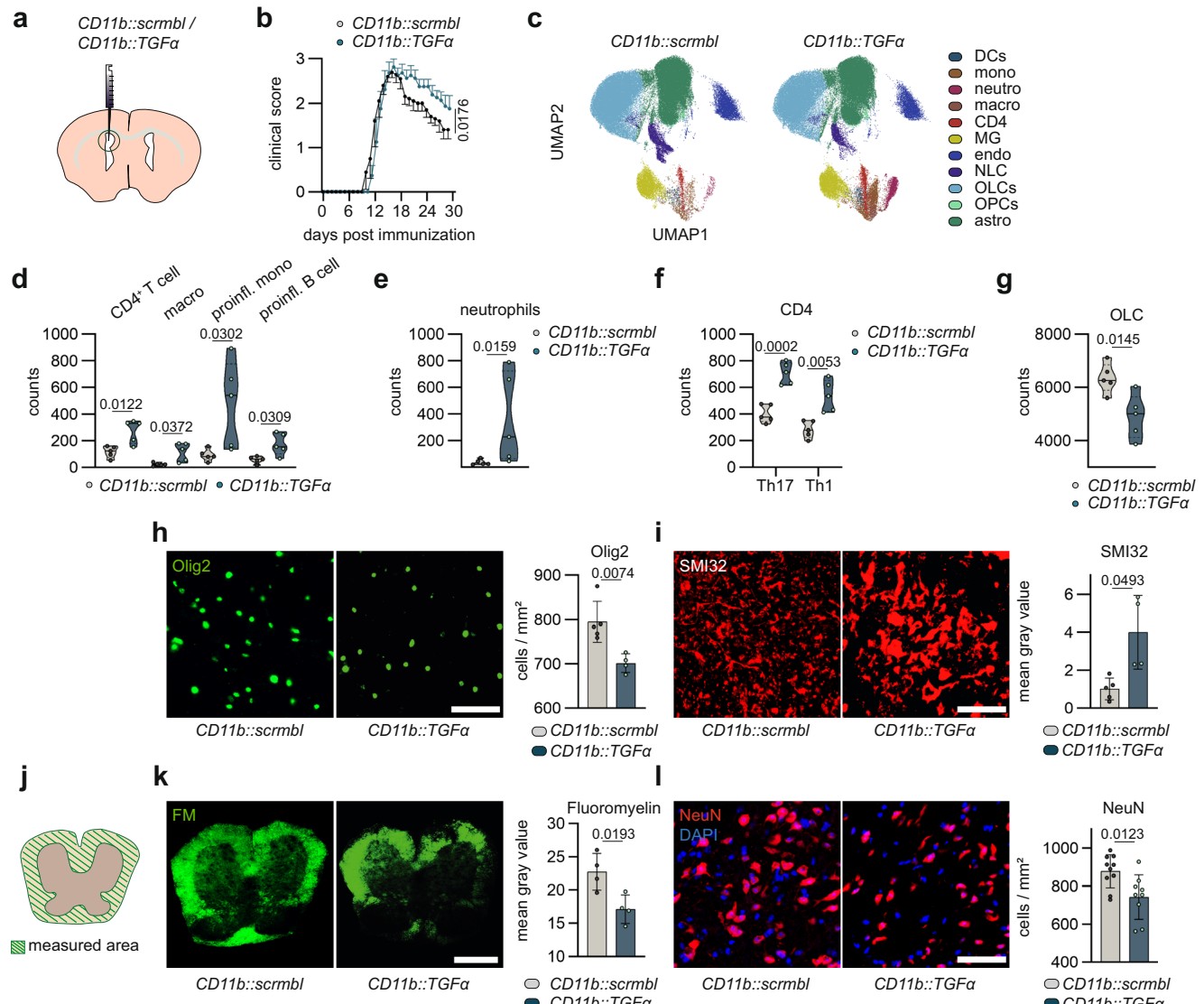

**Fig. 2 | Microglia-derived TGFα promotes protective effects during acute CNS inflammation. a** Delivery of lentiviruses (*CD11b::scrmbl; CD11b::TGFα*) via intracerebroventricular (i.c.v.) injection. **b** EAE development in mice transduced with *CD11b::scrmbl* (*n* = 5) or *CD11b::TGFα* (*n* = 5). **c** UMAP plot of CNS cells analysed by high-dimensional flow cytometry after dimensionality reduction in *CD11b::scrmbl* (*n* = 5) and *CD11b::TGFα* (*n* = 5) mice. **d** Abundance of CD4 + T cells, macrophages (macro), pro-inflammatory monocytes (mono) and B cells, neutrophils (**e**) oligodendrocyte lineage cells (OLC) (**g**) in the CNS of *CD11b::scrmbl* (*n* = 5) and *CD11b::TGFα* (*n* = 5) mice normalized to single live cells. **f** cytokine production by CD4 + T cells (IL17; IFNγ) in the CNS of *CD11b::scrmbl* (*n* = 5) and *CD11b::TGFα* (*n* = 5) mice analysed by intracellular flow cytometry. **h** Immunostaining (left) and quantification (right) of Olig2⁺ oligodendrocytes in spinal cords of *CD11b::scrmbl* (*n* = 5)

and *CD11b::TGFα* (*n* = 4) mice. Scale bar 50 μm. **i** Immunostaining (left) and quantification (right) of SMI32 in spinal cords of *CD11b::scrmbl* (*n* = 5) and *CD11b::TGFα* (*n* = 4) mice. Scale bar 50 μm. **j** Measured area (white matter) for immunostaining (left) and analysis (right) (**k**) of Fluoromyelin (FM) in spinal cords of *CD11b::scrmbl* (*n* = 5) and *CD11b::TGFα* (*n* = 4) mice. Scale bar 200 μm. **l** Immunostaining (left) and quantification (right) of NeuN⁺ neurons (red) and DAPI for nuclear staining in spinal cords of *CD11b::scrmbl* (*n* = 10) and *CD11b::TGFα* (*n* = 9) mice. Scale bar 50 μm. Analyses performed at day 30 post immunization. Data shown as mean ± SD. Data shown as mean ± SEM in (**b**). Two-tailed non-parametric Wilcoxin matched-pairs signed-rank test in (**b**). Two-way ANOVA with Sidak's multiple comparisons test in (**d**). Two-tailed unpaired *t*-test in (**e**–**l**).

proliferation markers within the peripheral immune compartment (Suppl. Fig. 4d–h), suggesting that TGFα treatment primarily acts within the CNS.

In our earlier investigations, TGFα emerged as key mediator orchestrating the interaction between microglia and astrocytes[13]. Next, to determine which effects identified in the present study were caused primarily by microglia to astrocyte interactions, we aimed to understand astrocyte-independent functions of TGFα. To that end, we knocked-out ErbB1 (EGFR) expression in astrocytes (*GFAP::Erbb1*, Suppl. Fig. 4i) before inducing EAE, and applied TGFα as outlined before. Indeed, even under astrocytic hyporesponsiveness to TGFα, its intranasal administration was still beneficial and reduced disease

(Fig. 4g), suggesting astrocyte-independent effects of TGFα in the CNS. This ameliorated clinical course was in line with a decline in IL17-producing monocytes and CD24⁺ CD4⁺ T cells (Suppl. Fig. 4j).

We next examined the therapeutic relevance of TGFα in clinically established disease, a setting relevant to MS patient care (Fig. 4a, h). Indeed, TGFα treatment starting after symptom onset promoted recovery from acute CNS inflammation by reducing CD44⁺ neutrophil numbers as well as the pro-inflammatory profile and proliferation of monocytes (Fig. 4i, j).

This was accompanied by an increase in OLC numbers in the CNS of TGFα-treated mice, along with an upregulation of OLIG2 expression in spinal cord tissue (Fig. 4k, l, Suppl. Fig. 4k). These observations were

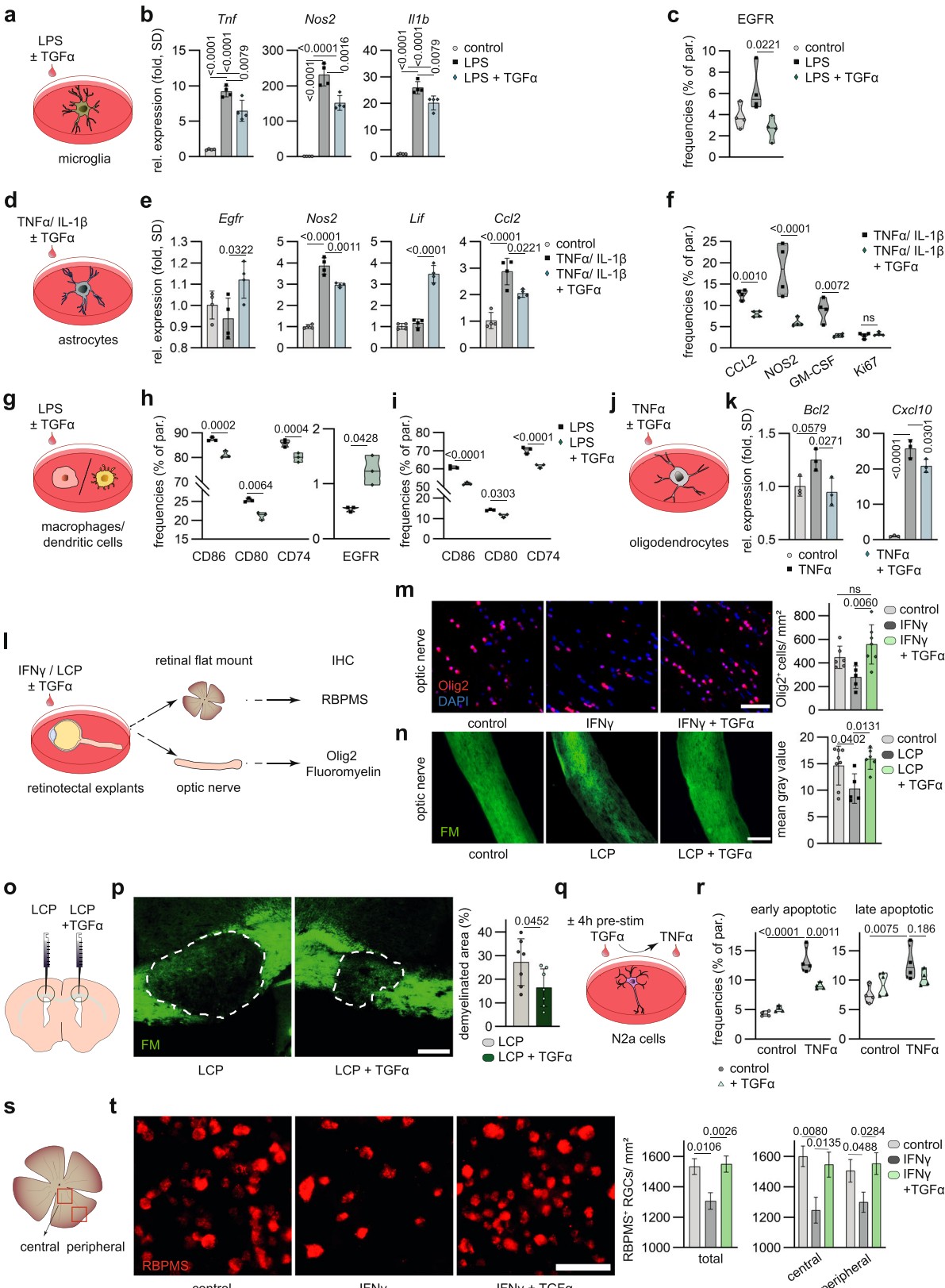

supported by a decrease in SMI32 in the spinal cord (Fig. 4m, Suppl. Fig. 4k) as well as increased myelinated white matter area (Fig. 4n). These effects were accompanied by increased survival of NeuN[+] neurons, suggesting a reduction in axonal damage, myelination, and increased neuronal survival following TGFα treatment (Fig. 4o, Suppl. Fig. 4k).

Previously, we have demonstrated up-regulation of TGFα in microglia in active MS lesions[13], consistent with our findings in EAE. To further investigate a still-prevailing deficit of TGFα, we assessed TGFα levels in cerebrospinal fluid (CSF) samples obtained from treatment-naïve patients diagnosed with relapsing-remitting multiple sclerosis (RRMS) and controls with non-inflammatory disease. Interestingly, we

**Fig. 3 | TGFα promotes protective effects under inflammatory and demyelinating conditions. a** Schematic and RT-qPCR analysis (**b**) of *Tnf*, *Nos2*, and *Il1b* expression in primary mouse microglia stimulated with LPS ± TGFα. *n* = 4 per group. **c** Abundance (% of parent) of EGFR in primary mouse microglia stimulated with LPS ± TGFα analysed by high-dimensional flow cytometry. *n* = 4 per group. **d** Schematic and RT-qPCR analysis (**e**) of *Egfr*, *Nos2*, *Lif* and *Ccl2* expression in primary mouse astrocytes stimulated with TNFα, IL-1β ± TGFα. *n* = 4 per group. **f** Relative expression (% of parent) of CCL2, NOS2, GM-CSF and Ki67 in primary mouse astrocytes stimulated with TNFα, IL-1β ± TGFα analysed by intracellular flow cytometry. *n* = 4 per group. **g** Schematic of bone-marrow derived macrophages (BMDM) and bone-marrow derived dendritic cells (BMDC) stimulated with LPS ± TGFα. *n* = 3/4 per group. **h** Relative expression (% of parent) of CD86, CD80, CD74 and EGFR in BMDM stimulated with LPS ± TGFα analysed by high-dimensional flow cytometry. *n* = 3 per group. **i** Flow cytometric analysis of CD86, CD80 and CD74 expression in bone-marrow derived dendritic cells (BMDC) stimulated with LPS ± TGFα. *n* = 3 per group. **j** Schematic and RT-qPCR analysis (**k**) of *Bcl2* and *Cxcl10* expression in primary mouse oligodendrocytes stimulated with TNFα ± TGFα. *n* = 3 per group. **l** Experimental setup for ex vivo culture of retinotectal system stimulated with IFNγ / lysolecithin (LCP) ± TGFα for immunohistochemical detection of RNA-binding protein with multiple splicing (RBPMS + ) retinal ganglion cells, Olig2+ oligodendrocytes and Fluoromyelin. **m** Immunostaining (left) and quantification (right) of Olig2+ oligodendrocytes and DAPI for nuclear staining in optic nerve explants following stimulation with IFNγ ± TGFα (− *n* = 5; + *n* = 7) or vehicle (*n* = 6). Scale bar 50 μm. **n**, Immunostaining (left) and quantification (right) of Fluoromyelin (FM) in optic nerve explants following stimulation with LCP ± TGFα (− *n* = 5; + *n* = 6) or vehicle (*n* = 8). Scale bar 200 μm. **o** Schematic of LCP-induced demyelination via injection of LCP ± TGFα into corpus callosum of each hemisphere. **p** Immunostaining (left) with Fluoromyelin (FM) and analysis (right) of LCP-induced demyelination in the corpus callosum. *n* = 7. Scale bar 200 μm. **q** Schematic and flow cytometric analysis (**r**) of neuronal cells (N2a) stained with Annexin V (A-V) and Propidium Iodid (PI) following stimulation with TNFα ± TGFα pre-stimulation of 4 hrs. **r** Quantification of early apoptotic (A-V + PI-) and late apoptotic (A-V + PI + ) neuronal cells following stimulation with TNFα ± TGFα, or vehicle. *n* = 4 per group. **s** Measured area (total, central, peripheral) for immunostaining (left) and quantification (right) (**t**) of RBPMS+ retinal ganglion cells in retinae stimulated IFNγ ± TGFα or vehicle (*n* = 4 per group). Scale bar 50 μm. Data shown as mean ± SD. Data shown as mean ± SEM in (**m–t**). One-way ANOVA with Tukey's multiple comparisons test in (**b–k**). One-way ANOVA with Dunnett's multiple comparisons test in (**m–t**). Two-way ANOVA with Sidak's multiple comparisons test in (**r**, **t**). Two-tailed unpaired in (**f–p**).

found reduced TGFα levels in the CSF of treatment-naïve RRMS patients as compared to controls (Fig. 4p), underlining the differential compartmentalization of tissue-intrinsic TGFα versus TGFα presented in the CSF, while suggesting a potential benefit of exogenously compensating a putative net deficiency of TGFα signaling in the CNS during acute inflammation. Indeed, TGFα-concentrations of CSF negatively correlated with clinical disability in Expanded Disability Status Scale (EDSS) (Fig. 4q). Of note, TGFα levels did not exhibit age-dependent variations (Suppl. Fig. 4l).

Taken together, TGFα plays multifunctional roles in CNS inflammation, leading to anti-inflammatory and tissue-protective effects. Intranasal administration of TGFα leads to its CNS penetration and limits the pro-inflammatory profiles of astrocytes and CNS infiltrating immune cells, while promoting neuronal survival and early oligodendrocyte precursor cell functions, thereby reducing lesion volumes and promoting recovery. Thus, TGFα represents a potential therapeutic target to promote lesion resolution in autoimmune CNS inflammation.

## Discussion

As a member of the epidermal growth factor (EGF) family, TGFα induces astrocyte proliferation and increases neuronal survival, migration and axonal growth in multiple contexts, including models of spinal cord injury (SCI) and ischemic stroke[18,28–31]. These neuroprotective effects are partially mediated by the induction of neurotrophic factor expression in astrocytes[28]. Considering the role of reactive astrocytes in promoting axon regeneration after SCI[32], it is reasonable to speculate that microglia-produced TGFα drives beneficial activities of astrocytes in the context of CNS diseases. Therefore, understanding the interactions between CNS resident and infiltrating immune cells in autoimmune CNS inflammation is crucial for developing effective therapeutic interventions for CNS-intrinsic neuroinflammation. Here, we integrated analyses of CSF samples from MS patients with preclinical mouse models to understand the role of TGFα in disease progression and resolution of autoimmune CNS inflammation. We identify microglia as the primary source of TGFα in the CNS during autoimmune inflammation, alongside limited and stage-specific expression of TGFα in astrocytes, oligodendrocytes, and neurons.

The relapsing-remitting phase of MS (RRMS) is mainly driven by the adaptive immune system, with peripherally primed adaptive immune cells (mainly T and B cells) crossing the BBB, causing inflammatory lesion formation and topologically defined neurological deficits[33]. Local anti-inflammatory and regenerative mechanisms limit this acute inflammation and promote tissue recovery, but often fail due to a preponderance of tissue-destructive mechanisms[33–35]. This study reveals the role of TGFα in promoting protective mechanisms during the recovery phase of acute CNS inflammation. Microglia-derived TGFα exhibits pleiotropic effects on various cell populations, including astrocytes, infiltrating T cells, pro-inflammatory myeloid cells, neurons and oligodendrocytes. Furthermore, through activation of the EGFR signaling pathway, TGFα mediates protective functions and contributes to the regulation of neuroinflammatory processes. Upon binding of a ligand, EGFR undergoes autophosphorylation and subsequent internalization, with the fate of either being dissociated or reintegrated into the membrane, depending on the activating ligand[36,37]. TGFα binding increases the probability of the EGFR being externalized to the cell surface, facilitated by TGFα dissociation in endosomes at low pH levels[38,39]. Previous studies have shown upregulated EGFR expression, especially in cancer cells, as a result of activated signaling pathways[40,41]. Furthermore, activation of this signaling pathway may influence EGFR gene expression[40,42], implying a regulatory role of TGFα in EGFR expression and forming a positive feedback loop. Our data suggest that TGFα, through EGFR activation, plays a dual role by both enhancing oligodendrocyte survival and modulating CD4+ T cell subsets, such as Th1, Th17, and GM-CSF-producing cells. TGFα signaling influences infiltrating immune cells, promoting their shift towards regulatory phenotypes through cytokine modulation. Similar to findings in a cerebral ischemia model[43], TGFα acts on oligodendrocytes by promoting their survival and preserving the integrity of axons, with the protective influence on OPCs being mediated by STAT3[29–31,43]. The EGFR signaling pathway activates multiple downstream cascades, including MAPK/ERK and PI3K/AKT, contributing to a balance between pro-inflammatory and reparative processes[41,44]. Consequently, the presence of TGFα favours the resolution of CNS inflammation, supporting previous research highlighting the neuroprotective properties of TGFα and its potential as a therapeutic agent[29–31,43]. In these lines, our study introduces the reduction of neutrophils and proliferating monocytes in response to intranasal TGFα treatment. While the pro-inflammatory functions of neutrophils in initiating inflammation have been well studied, their contribution in resolution of inflammatory processes, working with monocytes, has just been recognized recently, suggesting TGFα as a factor that might promote tissue protective properties of neutrophils[45–48]. In this study, we investigated TGFα's effects under both inflammatory and demyelinating conditions, both relevant in acute and late stages of MS[49,50]. Our findings indicate that recombinant TGFα regulates the pathogenic activities of various cell types, including astrocytes, macrophages, dendritic cells, oligodendrocytes, and neurons, as demonstrated in different experimental setups. This highlights the pleiotropic effects of

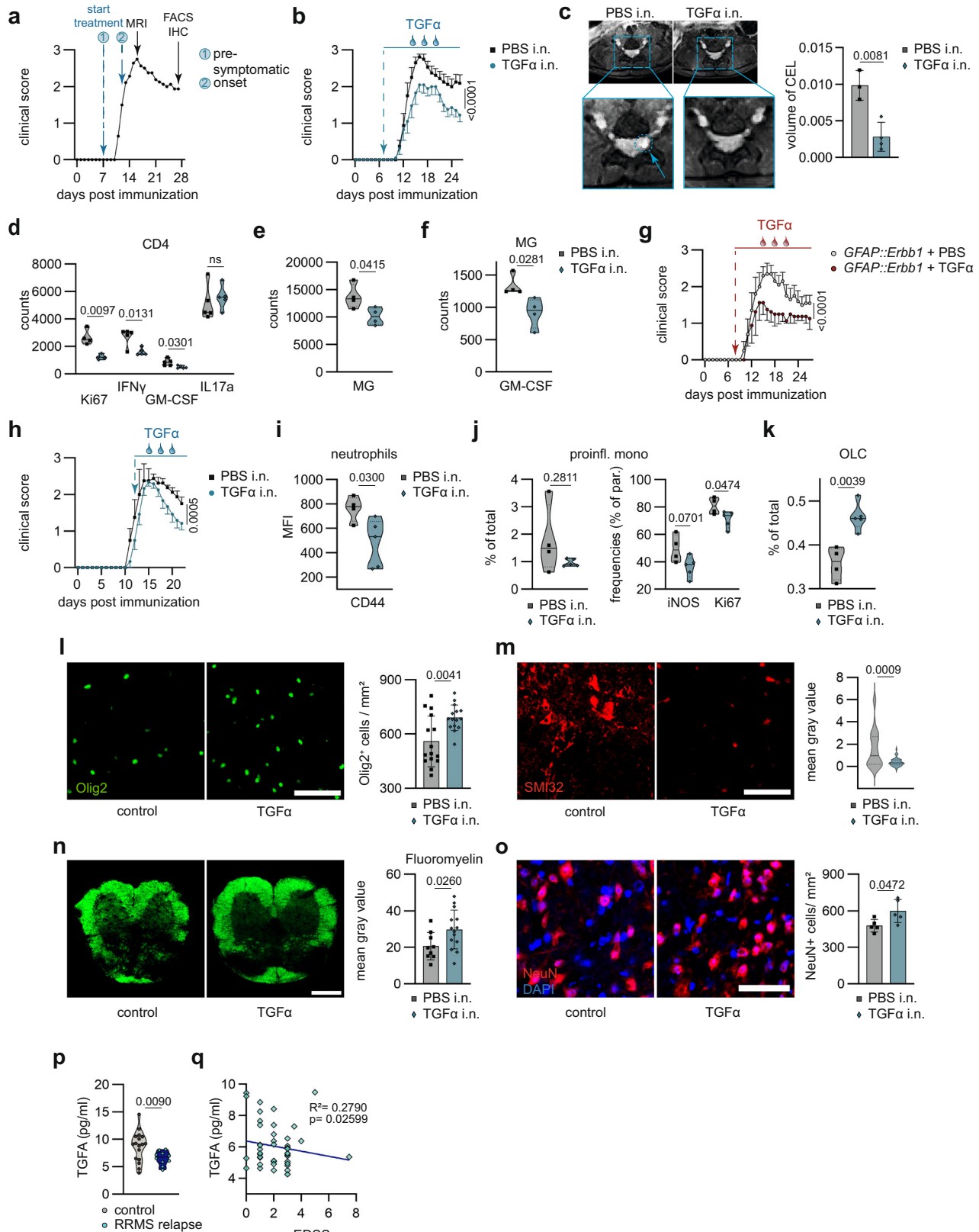

TGFα and suggests strategies for exploring its potential to mitigate both inflammatory and degenerative aspects of MS.

Apart from these observations, effective treatments specifically designed to promote resolution after inflammatory lesions in the CNS remain limited[11,12]. Current therapies, such as steroids, may alleviate acute symptoms, but they fail to change long-term disability outcomes

in MS patients[51,52]. In this context, intranasal application strategies using suitable agents such as TGFα might contribute to restore glia-mediated protective pathways to promote CNS regeneration. This delivery method allows for targeted application directly to the CNS, circumventing the challenges posed by the BBB[25,53]. In these lines, we here provide evidence that intranasal administration of

**Fig. 4 | TGFα as treatment target for lesion resolution in autoimmune CNS inflammation. a** Schematic workflow of intranasal TGFα treatment approach (pre-symptomatic and onset starting point) including MRI analysis at peak of disease and immunohistochemistry and flow cytometry during recovery from EAE. **b** EAE development in mice following intranasal treatment with TGFα or vehicle on a daily basis (start day 7) ($n = 8$). **c** MRI images (left) and quantification (right) of volume of contrast enhancing lesions (CEL, dotted line) in spinal cords of EAE mice following intranasal treatment with TGFα or vehicle. $n = 4$ per group. **d** CNS cells from TGFα or vehicle treated mice analysed by intracellular flow cytometry showing expression levels of proliferation (Ki67) marker and cytokine production (IFNγ, GM-CSF, IL17a) in CD4 + T cells. **e** Abundance of microglia (MG) including microglial GM-CSF production (**f**) in the CNS of TGFα or vehicle treated mice. $n = 4$ per group. **g** EAE development in mice transduced with *GFAP::scrmbl* ($n = 5$) *GFAP::Erbb1* ($n = 4$) following intranasal treatment with TGFα or vehicle on a daily basis (start day 7). **h** EAE development in mice following symptomatic intranasal treatment with TGFα ($n = 6$) or vehicle ($n = 4$) on a daily basis (start day 12). **i** Abundance of CD44+ neutrophils (MFI) and pro-inflammatory monocytes (**j** left; % of total) and their cytokine (iNOS, Ki67) production (**j** right; % of parent) in the CNS of TGFα or vehicle treated mice (symptomatic). $n = 4/5$ per group. **k** Abundance of OLCs (% of total) in the CNS of TGFα or vehicle treated mice (symptomatic). $n = 4/5$ per group. **l** Immunostaining (left) and quantification (right) of Olig2+ oligodendrocytes in the spinal cord in EAE mice following symptomatic intranasal treatment with TGFα. $n = 14/15$ per group. Scale bar 50 μm. **m** Immunostaining (left) and quantification (right) of SMI32 in spinal cords of EAE mice following symptomatic intranasal treatment with TGFα. Scale bar 50 μm. **n** Immunostaining (left) and analysis (right) of Fluoromyelin (FM) in spinal cords following symptomatic intranasal treatment with PBS ($n = 9$) or TGFα ($n = 14$). Scale bar 200 μm. **o** Immunostaining (left) and quantification (right) of NeuN+ neurons and DAPI for nuclear staining in spinal cords of EAE mice following symptomatic intranasal treatment with TGFα. Scale bar 50 μm. $n = 5$ per group. **p**, Enzyme-linked Immunosorbent Assay (ELISA) of cerebrospinal fluid (CSF) samples from individuals with relapsing-remitting multiple sclerosis (RRMS; $n = 10$) and healthy controls ($n = 17$). **q** Simple linear regression analysis of Multiplex analysis (Linnerbauer et. al., 2024) of cerebrospinal fluid (CSF) samples from individuals with relapsing-remitting multiple sclerosis (RRMS; $n = 47$) and healthy controls ($n = 20$). Data shown as mean ± SD. Data shown as mean ± SEM in (**b**–**h**). Two-tailed non-parametric Wilcoxin matched-pairs signed-rank test in (**b**– **h**). Two-way ANOVA with Sidak's multiple comparisons test in (**d**, **j** right). Two-tailed unpaired *t*-test in (**c**– **j** left, **k**–**p**). Simple linear regression in (**q**).

CNS-penetrating TGFα reduces the number and polarization of CNS infiltrating immune cells, limiting tissue-destructive CNS inflammation in preclinical model systems. Consistent with previous studies employing TGFα as a therapeutic approach in stroke mouse models[31,43], our study suggests a therapeutic potential of TGFα to mitigate CNS inflammation worthy of further studies.

Several limitations of our study must be taken into account. First and foremost, this is a preclinical study and translation to human disease must be evaluated in future preclinical and potentially clinical studies. Indeed, even though we did not detect immunological alterations in the peripheral immune compartment upon intranasal TGFα application, proliferation-inducing as well as yet unidentified effects of TGFα may raise concerns about potential side effects upon its intranasal application. Thus, future studies should address both peripheral and CNS-intrinsic effects of nasally applied TGFα and extend investigations into diverse models, different time points of sampling and additional preclinical and clinical settings to ensure the safety of this approach. Next, we have previously demonstrated enhanced TGFα expression in human MS tissue during acute autoimmune inflammation[13]; by contrast, we here detected reduced levels of TGFα in the CSF. Our study cohorts were not age-matched ideally, but a 7 year difference between controls and MS patients was notable. Even though we did not detect age-dependency of TGFα levels in CSF, future studies must include both cross-sectional and, where feasible, longitudinal analyses of TGFα levels in the CSF to rule out hence unidentified age-dependent concentration changes of TGFα. For now, we suggest that specific compartmentalization of TGFα between tissue and CSF might be the most plausible explanation of our observations, which might also highlight a potential therapeutic use of correcting a net lack of protective TGFα signaling in the CNS. Finally, we focused on the specific role of TGFα in autoimmune CNS inflammation. However, its potential interactions with TGFβ, known for its immune-modulating effects[54–63], as well as with other proteins and agents is warranted, to rule out cross-reactions of EGFR-targeting approaches. Another open question is whether TGFα modulates specific CD8+ T cell subsets involved in disease resolution. In particular, regulatory CD122+ CD8+ T cells have been associated with the remission phase of EAE, yet their role in the context of TGFα-mediated immune modulation remains unexplored[64]. Future studies should investigate this potentially relevant effector pathway. Finally, another limitation of our study is that we did not fully explore the entire spectrum of TGFα-mediated effects; for instance, the impact of TGFα on stromal cells, which are critical for blood-brain barrier integrity and immune surveillance, remains unexamined. Additionally, there is a possibility of tolerance development with prolonged treatment. These concerns should be addressed in future preclinical studies to ensure a comprehensive evaluation of the therapeutic strategy.

In summary, this study demonstrates the spatio-temporal regulation of TGFα in autoimmune CNS inflammation. We identify its pleiotropic effects on different cell populations during autoimmune CNS inflammation. Microglia-derived TGFα emerges as a regulator of immune checkpoints in both CNS resident and infiltrating immune cells, preserving oligodendrocyte function, and improving neuronal survival, thereby promoting resolution of inflammation in the CNS. Given the relative reduction of TGFα in the CSF of MS patients, alongside its negative correlation with disability at the time of relapse, BBB-penetrating intranasal delivery might represent an approach to target CNS-intrinsic autoimmune inflammation. In future studies, evaluation of intranasal application of CNS-penetrating TGFα may represent a train of research to modulate CNS-intrinsic processes relevant to autoimmune inflammation.

## Methods
These studies comply with the relevant ethical regulations.

### Mice
*C57Bl/6 J* mice were obtained from Jackson Laboratory (#000664) and housed in groups of two–five per cage with ad libitum access to water and food under a standard light cycle (12 h:12 h light:dark) (lights from 07:00 am to 07:00 pm) at 20-23 °C and humidity ( ~ 50%). Female mice aged 8–12 weeks and pups (P0–P6) were used on a *C57Bl/6 J* background. Mice were euthanized by transcardial perfusion under isoflurane anesthesia according to the approved protocols.

### EAE
Female mice (8–12 weeks) were immunized by subcutaneous injection of 150 μg of myelin oligodendrocyte glycoprotein (MOG)$_{35-55}$ peptide (Genemed Synthesis, #MOG3555-P-5) emulsified in freshly prepared complete Freund's adjuvant (CFA), which was made from IFA (BD Biosciences, BD263910) plus *Mycobacterium tuberculosis* H-37Ra (BD Biosciences, BD231141) at a 1:1 ratio. Mice were injected intraperitoneally twice with 200 ng of pertussis toxin (PTX) (List Biological Laboratories, LBL180) in 200 μl PBS on the day of immunization and 2 days later. All mice were monitored and scored on a daily basis thereafter by blinded investigators. EAE timepoints were defined as: disease onset (day 13), peak (day 17), recovery (day 24), late recovery (day 34), and extremely late recovery (day 56). The EAE clinical scores were defined as follows: 0, no signs; 1, fully limp tail; 2, hindlimb

weakness; 3, hindlimb paralysis; 4, forelimb paralysis; and 5, moribund. To ensure blinding, animals were coded by a lab member not involved in scoring. EAE experiments were performed in at least three independent cohorts ($n \geq 3$), with each group comprising a minimum of five animals ($n \geq 5$).

## Lentivirus production

The lentiviral vectors were created following established procedures[19,65–67]. Specifically, the lentiviral vectors were obtained from lentiCRISPRv2 (Addgene #52961[68]) and lentiCas9-EGFP (Addgene #63592[69]). To generate CRISPR/Cas9 lentiviral constructs, modifications were made to the pLenti-U6-sgScramble-Gfap-Cas9-2A-EGFP-WPRE lentiviral backbone. The *Gfap* promoter used in this study was the ABC$_1$D *gfa2 GFAP* promoter, and the *Itgam* promoter (also known as Cd11b) was described previously[13]. The substitution of sgRNAs was achieved through a three-way cloning strategy using specific primers (U6-PCR-F 5′-AAAGGCGCGCCGAGGGCCTATTT-3′, U6-PCR-R 5′-TTTTTTGGTCTCCCGGTGTTTCGTCCTTTCCAC-3′, cr-RNA-F 5′-AAA AAAGGTCTCTACCG(sgRNA)GTTTTAGAGCTAGAAATAGCAAGTT-3′, cr-RNA-R 5′-GTTCCCTGCAGGAAAAAAGCACCGA-3′). Phusion Master Mix (Thermo Fisher Scientific, #F548S) was used to amplify the products, which were then purified with the QIAquick PCR Purification Kit (Qiagen, 28104). Digestions were carried out using DpnI (NEB, #R0176S), BsaI-HF (NEB, #R3535/R3733), AscI (NEB, #R0558), or SbfI-HF (NEB, #R3642). The ligated constructs were transformed into NEB Stable Cells (NEB, #C3040), and single colonies were selected. Plasmid DNA was isolated using the QIAprep Spin Miniprep Kit (Qiagen, #27104). The lentiviral plasmids were then transfected into HEK293FT cells following the ViraPower Lentiviral Packaging Mix protocol (Thermo Fisher Scientific, #K497500) along with pLP1, pLP2, and pseudotyped with pLP/VSVG. After changing the medium the next day, the lentivirus was collected 48 h later and concentrated using the Lenti-X Concentrator (Clontech, #631231) as per the manufacturer's protocol. The concentrates were resuspended in 1/100 of the original volume in PBS. For the delivery of lentiviruses via intracerebroventricular (i.c.v.) injection, the procedure described in refs. 19,65,67 was followed. Briefly, mice were anesthetized with a mixture of 1% isoflurane and oxygen. The mice's heads were shaved and cleaned using 70% ethanol and Lidocain-gel. A medial incision was made in the skin to expose the skull, and the ventricles were targeted using specific coordinate relative to Bregma: +1.0 (lateral), −0.44 (posterior), −2.2 (ventral). The lentivirus was injected into the mice via 5 µl injections using a 10 µl Hamilton syringe (Sigma-Aldrich, #20787) on a Stereotaxic Alignment System (Kopf, #1900). The mice were then sutured and allowed to recover in a separate clean cage. Subcutaneous injection of Meloxicam (3 mg/kg) were administered every 12 h post i.c.v. injection for a period of 2 days. The mice were given 4–7 days to recover before the induction of EAE. The CRISPR-Cas9 sgRNA sequences were designed using a combination of the Broad Institute's sgRNA GPP Web Portal and Synthego. The specific sgRNAs used in this study were Tgfa 5′-TGCCCCAGCAGACTCACCCG-3′, Erbb1 5′-GATGT ACAACAACTGTGAAG-3′, Scrmbl 5′-GCACTACCAGAGCTAACTCA-3′. For knockout validation, microglia (Ly6C$^-$CD11b$^+$) were isolated from the CNS of EAE mice using the mouse anti-Ly6C-Biotin (Miltenyi, #130-111-914), anti-Biotin MicroBeads UltraPure (Miltenyi, #130-105-637), and the CD11b Microbead Isolation kit (Miltenyi, #130-049-601) according to the manufacturer's instructions. Knockout validation at protein level is provided in the supplementary material.

## Intranasal TGFα treatment

Intranasal delivery of recombinant human transforming growth factor alpha (TGFα) (PeproTech, 100−16 A) or vehicle control (PBS) was applied in 10 µl dropwise to each nostril. Daily treatment with 5.5 µg TGFα dissolved in PBS per mouse was started on day 7 following EAE

induction or at the onset of disease (day 11). Mice were monitored and scored as described previously.

## Primary glial cell culture experiments

To isolate primary mixed glial cells, we performed experiments using P0-P4 pups from the *C57Bl/6J* strain following established protocols[13,25,26]. The brains were carefully dissected in ice-cold PBS, and 6–8 brains were pooled. After centrifugation at 500 g for 10 min at 4 °C, the resulting pellet was resuspended in a 0.25% Trypsin-EDTA (Thermo Fisher Scientific, #25200-072) solution and incubated at 37 °C for 10 min. To enhance the digestion process, DNase I (Thermo Fisher Scientific, #90083) was added, and the brains were further incubated for an additional 10 min at 37 °C. At the end of incubation, warm growth medium consisting of DMEM + GlutaMAX (Thermo Fisher Scientific, #61965026) supplemented with 10% FBS (Thermo Fisher Scientific, #10438026) and 1% penicillin/streptomycin (Thermo Fisher Scientific, #10500064) was added to stop trypsinization. The cell suspension was then filtered through a 70 µm cell strainer and subjected to centrifugation at 500 g for 10 min at 4 °C. The resulting cell pellet was resuspended in DMEM + GlutaMAX supplemented with 10% FBS and 1% penicillin/streptomycin, and the cell solution was transferred to T-75 flasks (Sarstedt, #83.3911.002) that had been coated with Poly-L Lysine (PLL, Provitro, #0413) at a concentration of 2 µg/ml. The flasks were incubated in a humidified incubator set at 37 °C with 5% CO2 for a period of 5–7 days until the cells reached confluency. The growth medium in the flasks was exchanged every 2–3 days to ensure cell viability.

To isolate a purified microglial and astrocytic population, the mixed glial cells were subjected to shaking at 180 rpm for 30 min. The supernatant was collected, the growth medium was replaced, and the cells were then shaken at a higher speed of at least 220 rpm for a minimum of 2 h. The resulting supernatant was collected, and the medium was replaced once again. Adherent cells (astrocytes) were detached from the culture flasks using TrypLE (Thermo Fisher Scientific, #12604013). CD11b$^+$ microglia were isolated from the collected supernatant and the shake-off fraction using the CD11b Microbead Isolation kit (Miltenyi, #130-049-601) according to the manufacturer's instructions. For subsequent stimulation experiments, both astrocytes (CD11b$^-$ adherent cells) and microglia were seeded into 48-well plates (Sarstedt, #NC1787625) that had been coated with PLL at a density of 150,000 cells per well. For flow cytometric analyses, cells were seeded into 24-well plates (TPP /M&B Stricker, #92024) at a density of 300,000 cells per well. Cell purity was confirmed by flow cytometry, as shown in Suppl. Fig. 3a.

## Primary mouse OPC cultures

Isolation of OPCs and differentiation experiments were conducted as described previously[58]. To culture primary oligodendrocyte precursor cells (OPCs), we utilized P0-P2 pups and performed brain dissection. The tissue dissociation and culturing followed the protocol used for primary glial cell culture experiments. Upon obtaining confluency of the mixed glial culture after 8–14 days, OPCs were isolated using O4 (Miltenyi, #130-094-543) and CD140a (Miltenyi, #130-101-547) microbead isolation kits as instructed by the manufacturer. The separated OPCs were then transferred to PLL and fibronectin (Sigma Aldrich, # F1141-1MG, 1 µg/ml) precoated T-75 flasks. The cells were maintained under proliferation conditions oligodendrocyte culture medium containing DMEM/F12 + GlutaMAX (Thermo Fisher Scientific, #31331-093), 1% penicillin/streptomycin (Thermo Fisher Scientific, #15140122), B27 Supplement (Thermo Fisher Scientific, #12587-010), N2 Supplement (Thermo Fisher Scientific, #17504-044), recombinant human basic fibroblast growth factor (bFGF, R&D, #233-FB) at a concentration of 20 ng/ml, and 10 ng/ml recombinant rat platelet-derived growth factor-AA (PDGF-AA; R&D, #1055-AA-050) for up to 6 days until

confluency. Subsequently, the cells were transferred to either 24-well or 48-well plates in preparation for the stimulation experiments.

For the differentiation of OPCs into myelinating oligodendrocytes, we utilized P0-P2 pups and dissected the forebrain according to a modified version from[70]. The samples were collected in a rotating C tube with 1 ml DMEM (PAN Biotech, P04-05410), placed into a gentleMACS™ Octo Dissociator with heaters and mechanically dissociated. After dissociation, mixed glial cells were seeded onto T75 flasks pre-coated with Poly-D-lysine hydrobromide (PDL; Sigma-Aldrich, #P6407) at 5 µg/ml. The cell suspension containing mixed glial cells were cultured for 7 days at 37 °C and 5% CO$_2$, with a change of medium every second day: DMEM supplemented with 20% FBS and 1% penicillin/streptomycin (Thermo Fisher Scientific, #15140122). To isolate oligodendrocyte precursor cells (OPCs), flasks containing mixed glial cells were subjected to an orbital shaker at 200 rpm and 37 °C for 18 h. The supernatant containing loosely microglia and OPCs was collected in a petri dish and incubated for 20 min at 37 °C to allow microglial attachment. Subsequently, remaining OPCs were seeded into cell culture plates coated with PDL- and laminin (1 µg/ml, Roche, #11243217001). Cell purity was confirmed by flow cytometry (Suppl. Fig. 3). The purified cells were treated with medium containing DMEM/F12 + GlutaMAX (Thermo Fisher Scientific, #31331-093), 1% penicillin/streptomycin, B27 Supplement (Thermo Fisher Scientific, #12587-010), and N2 Supplement (Thermo Fisher Scientific, #17504-044). During the initial 24 h, 10 ng/ml recombinant rat platelet-derived growth factor-AA (PDGF-AA; R&D, #1055-AA-050) and 10 ng/ml recombinant human basic fibroblast growth factor (bFGF; R&D, #233-FB) were additionally supplemented. The next day, proliferation medium was replaced with fresh oligodendrocyte culture medium. Different conditions were employed, including the addition of 50 ng/ml recombinant human TGFα (R&D, #239-A-100), 10 ng/ml PDGF-AA/bFGF as positive control for OPCs, or 40 ng/ml 3,3′,5-Triiodo-L-thyronine (T3) as positive control for differentiation into myelinating oligodendrocytes. To assess the differentiation of oligodendrocyte lineage cells, flow cytometry and RT-qPCR analyses were conducted after.

### Isolation of mouse bone marrow macrophages and dendritic cells

Experiments followed established procedures[58]. Bone-marrow progenitor cells were harvested from the femurs and tibias of *C57Bl/6 J* mice. In brief, bones were cleared of tissue, briefly washed in 70% ethanol, and rinsed in PBS. Bone marrow was flushed with PBS, gently mixed and centrifuged at 400 g for 10 min at 4 °C. After red blood cell lysis with ACK lysing buffer (Life Technology, #A10492-01), the cell suspension was filtered and seeded onto 10 cm dishes in DMEM + GlutaMAX (Thermo Fisher Scientific, #31331-093) containing 1% penicillin/streptomycin (Thermo Fisher Scientific, #15140122) and 10% FBS (Thermo Fisher Scientific Scientific, #10438026). For differentiation into BMDMs, 20 ng/mL macrophage colony-stimulating factor (M-CSF; Peprotech, #315-02) was added to the culture medium, while granulocyte-macrophage colony-stimulating factor (GM-CSF; 20 ng/mL, Peprotech, #315-03) was employed for BMDC cultures. Cultures were maintained for 7–10 days, with media changes on days 4 and 6. Cells were detached with TrypLE for further analysis once confluency and characteristic morphology were confirmed.

### Cell culture experiments and stimulants

Unless specified otherwise, the concentrations listed below were utilized for the stimulation experiments with different molecules: LPS-EB (InvivoGen, #tlrl-3pelps) 10 ng/ml, recombinant mouse TNFα (Peprotech, #AF-315-01A) 10 ng/ml, recombinant mouse IL-1β (Peprotech, #211-11B) 5 ng/ml, recombinant human TGFα (R&D, #239-A-100) 20 ng/ml, IFN-γ (R&D, #485-MI-100/CF) 100 ng/ml. To conduct RT-qPCR

analysis, the seeded cells were incubated for a duration of 24 h. For flow cytometry, the cells were incubated for a period of 48 h.

### Neurotoxicity assay

Neuro2a cells (N2a, American Type Culture Collection, CCL-131) were seeded at a density of 200.000 cells per well in a 24-well plate for the apoptosis assay. The cells were pre-stimulated with TGFα (50 ng/ml) for 4 h before inducing apoptosis with TNFα (20 ng/ml) for 48 h. Unstimulated and TNFα-stimulated growth media were used as controls. The cells were detached after stimulation, transferred to a 96U well plate, and washed once with PBS. The cells were then stained with LIVE/DEAD Fixable Aqua Dead Cell Stain Kit (Thermo Fisher Scientific, L34957) according to the manufacturer's instructions. Furthermore, we conducted annexin V-propidium iodide staining using the APC Annexin V Apoptosis Detection Kit with PI (Biolegend, 640932) following the manufacturer's instructions. The cells were washed once and suspended in annexin V binding buffer prior to being acquired on a 3 L Cytek Northern Lights flow cytometer. Flow cytometry data analysis was carried out using the OMIQ platform.

### Ex vivo optic nerve and retina culture

We used mice with a *C57Bl/6 J* background for ex vivo optic nerve and retina tissue culture. After euthanasia, the eyes, including the optic nerves attached to the eyeball, were collected and immediately placed in ice-cold PBS. The eye was further opened along the ciliary body as the residual tissue was detached. The cornea, vitreous body, and lens were removed, and the optic nerve–retinal unit was directly transferred into the culture wells of the experimental setup. The culture medium for the optic system consisted of 50% Opti-MEM, 25% Hank's Balanced Salt Solution (HBSS), 25% fetal calf serum (FCS), 25 mM D-glucose and penicillin-streptomycin solution. An unstimulated control, IFNγ (100 ng/ml) and IFNγ + TGFα (50 ng/ml) stimulation were used in the experimental layout. An additional setup was performed by substituting LCP (40 µg/ml) for IFNγ. For each condition, four optic nerve–retinal units were pooled and incubated in the appropriate medium for 24 h. The retinae were dissected at the end of the culture period for immunofluorescence retinal flat mount analysis to quantify ganglion cells. The optic nerve was separated from the retina, and the retinal pigment epithelium was removed from the retina. Retinal tissue was incised four times and placed in a 24-well plate filled with ice-cold 1x PBS. Retinae were frozen in fresh 2% TritonX-100 in 1x PBS at -80 °C after one washing step with 2% TritonX-100 in 1x PBS at room temperature. After a washing step, the remaining optic nerves were fixed with 4% paraformaldehyde (PFA) for 30 min at room temperature. Subsequently, the tissue was washed, embedded in TissueTec, and frozen in nitrogen-cooled methyl butane. Tissue samples were frozen and stored at -80 °C until further processing. Longitudinal cryosections of the optic nerves with a thickness of 10 µm were then obtained on glass slides.

### Lysolecithin induced demyelination of the corpus callosum

Demyelination in the corpus callosum was induced using Lysolecithin (LCP), following a previously described protocol[71]. Briefly, a 1% LCP solution was prepared by mixing lysolecithin from Sigma Aldrich (#L4129) with sterile 1× PBS. For the LPC + TGFα injection, a 1% LPC solution containing 100 ng of recombinant human TGFα (PeproTech, 100-16 A) was prepared in 1x sterile PBS. To administer LCP into the corpus callosum, mice were anesthetized using 1% isoflurane mixed with oxygen. Their heads were shaved and cleaned with 70% ethanol and Lidocaine-gel. A medial incision was made on the skin to expose the skull. The corpus callosum was targeted bilaterally with the following coordinates relative to Bregma: ±1.0 (lateral), +1.3 (anterior), −2.5 (ventral). The contralateral side served as an internal control. Mice were injected with 2 µl of either vehicle (PBS) or LPC ± TGFα using a 10 µl Hamilton syringe from Sigma-Aldrich (#20787) on a Stereotaxic

Alignment System (Kopf, #1900). After the injection, the incision was sutured, and mice were allowed to recover in a separate clean cage. To minimize discomfort, mice received a subcutaneous injection of 1 mg/kg Meloxicam after the intracerebroventricular (i.c.v.) injection and again 48 h later. Analysis of the mice was performed 7 days post-injection.

## Isolation of cells from adult CNS

To isolate the cells from the CNS, mice were euthanized and perfused with cold 1× PBS. Their CNS were isolated and mechanically diced using sterile razors. Brains and spinal cords were processed separately, unless otherwise specified, and placed in a 5 ml enzyme digestion solution consisting of 35.5 μl papain suspension (Worthington, #LS003126) diluted in enzyme stock solution (ESS), and then equilibrated to 37 °C. The ESS was prepared with the following components: 10 ml 10× EBSS (Sigma-Aldrich, #E7510), 2.4 ml 30% D(+)-glucose (Sigma-Aldrich, #G8769), 5.2 ml 1 M NaHCO3 (VWR, #AAJ62495-AP), 200 μl 500 mM EDTA (Thermo Fisher Scientific, #15575020), and 168.2 ml ddH2O, filter-sterilized through a 0.22-μm filter. The samples were then shaken at 80 rpm for 30–40 min at 37 °C to facilitate enzymatic digestion. Digestion was halted by adding 1 ml of 10× hi ovomucoid inhibitor solution and 20 μl of 0.4% DNase (Worthington, #LS002007) diluted in 10 ml inhibitor stock solution (ISS). The 10× hi ovomucoid inhibitor stock solution contained 300 mg BSA (Sigma-Aldrich, #A8806) and 300 mg ovomucoid trypsin inhibitor (Worthington, #LS003086) diluted in 10 ml 1× PBS and filter-sterilized using a 0.22 μm filter. The ISS contained 50 ml 10× EBSS (Sigma-Aldrich, #E7510), 6 ml 30% D(+)-glucose (Sigma-Aldrich, #G8769), and 13 ml 1 M NaHCO3 (VWR, #AAJ62495-AP) diluted in 170.4 ml ddH2O and filter-sterilized through a 0.22 μm filter. The homogenized tissue was then mechanically dissociated using a 5 ml serological pipette and triturated through a 70 μm cell strainer (Fisher Scientific, #22363548) into a fresh 50 ml conical tube. Following this, the tissue was centrifuged at 600 g for 5 min and resuspended in 10 ml of 30% Percoll solution (9 ml Percoll from GE Healthcare Biosciences, #17-5445-01, 3 ml 10× PBS, and 18 ml ddH2O) for purification. The Percoll suspension was centrifuged at 600 g for 25 min with no brakes. After discarding the myelin top layer and supernatant, the remaining mononuclear cell pellet was washed once with 1× PBS, centrifuged at 500 g for 5 min, and prepared for further applications.

## Isolation of splenic cells

The spleens were mechanically dissected, and the tissue was further dissociated by triturating it through a 100-μm cell strainer (Thermo Fisher Scientific, #10282631). To remove red blood cells, an ACK lysing buffer (Life Technology, A10492-01) was applied for 5 min. Subsequently, the cells were washed with a solution containing 0.5% BSA and 2 mM EDTA at pH 8.0 in 1× PBS. After this process, the cells were prepared for use in further applications.

## Flow cytometry

For the characterization of splenic and CNS cells we used flow cytometric analyses. The LIVE/DEAD™ Fixable Aqua Dead Cell Stain Kit (Thermo Fisher Scientific, #L34957) was used to identify viable cells according to the manufacturer's instructions. After washing the cells in PBS, they were labelled with flow cytometry antibodies at 4 °C in the dark for 30 min, diluted in FACS buffer (1x PBS, 2% FBS, 2 mM EDTA). The cells were then washed twice with FACS buffer and resuspended in 1× PBS for acquisition. The following antibodies were used in the study: BV421-CD11b (Biolegend, #101235), BV480-CD11c (BD, #565627), BV510-F4/80 (Biolegend, #123135), BV570-Ly6C (Biolegend, #128029), BV605-CD80 (BD, #563052), BV650-CD56 (BD, #748098), BV650-CD8 (BD, #100741), PE-eFlour610-CD140a (Thermo Fisher Scientific, #61140180), SuperBright780-MHCII (Thermo Fisher Scientific, #78532080), BV711-CD74 (BD, #740748), AF488-EGFR (Cell Signaling,

#5616), PE-Cy7-Tmem119 (Thermo Fisher Scientific, #25611980), eFluor 450-CD3 (Thermo Fisher Scientific, #48003742), BV605-CD68 (Biolegend, #137021), BV711-CD74 (BD, #740748), SB600-CD140B (Thermo Fisher Scientific, #63140282), PE-CD105 (Thermo Fisher Scientific, #12-1051-82), AF488-A2B5 (Novus Biologicals, #FAB1416G), PE-Cy5-CD24 (Biolegend, #101811), PE-Cy7-CD31 (Thermo Fisher Scientific, #25031182), PerCP-eFlour710-CD86 (Thermo Fisher Scientific, #46086280), AF532-CD44 (Thermo Fisher Scientific, #58044182), PE-B220 (BD, #561878), PE-ACSA1 (Miltenyi Biotech, #130118483), PE-Cy5.5-CD45 (Thermo Fisher Scientific, #35045180), JF646-MBP (Novus Biologicals, #NBP2-22121JF646), APC-Cy7-Ly6G (Biolegend, #127623), AF700-O4 (R&D, #FAB1326N), BUV737-CD154 (BD, #741735), AF660-CD19 (Thermo Fisher Scientific, #606019380), APC/Fire810-CD4 (Biolegend, #100479), PE-ACSA2 (Miltenyi Biotech, #130117386).

For intracellular flow cytometry staining, cells were fixed overnight after surface staining using the eBioscience™ Foxp3 / Transcription Factor Staining Buffer Set (eBioscience, #00552300) as per the manufacturer's instructions. The following antibodies were used for intracellular cytokine staining: PE-eFlour610-iNOS (eBioscience, #61592080), BV711-IL17a (Biolegend, #506941), PE-Cy5-FoxP3 (Thermo Fisher Scientific, #15-5773-82), PE-Cy7-IFNγ (Biolegend, #505826), PE-CCL2 (BD, #554443), PE PerCP-eFlour710-TNF (eBioscience, #46732180), APC-GM-CSF (eBioscience, #17733182), APC-eF780-Ki67 (Thermo Fisher Scientific, #506941), PE-IL10 (BD, #554467), JF646-TGFα (Novus Biologicals, #NBP2-34683JF646), AF700-Ki67 (Thermo Fisher Scientific, #56569882). The rabbit anti-mouse phospho-EGFR (Tyr1068; Thermo Fisher Scientific; #44-788 G) antibody was conjugated with the APC Conjugation Kit - Lightning-Link® (Abcam; #ab201807) after manufacturer's instructions and used for intracellular flow cytometry staining. Samples were blinded with non-descriptive codes to ensure objective data analysis.

## Analysis of multiparameter flow cytometry data

The OMIQ platform was utilized for data analysis. Briefly, cells were gated according to previously described methods[66,72]. Microglia were identified by: singlet gating, followed by CD45$^{int}$CD11b$^+$ expression and the absence of Ly6C and Ly6G; CNS macrophages: singlet gating, CD45$^{high}$CD11b$^+$, MHCII$^+$, Ly6C$^-$, CD11c$^-$. Detailed gating strategy in Suppl. Fig. 5. For dimensionality reduction, cells were downsampled to an appropriate number per group. Opt-SNE was performed (maximum 1000 iterations, perplexity 30, theta 0.5, verbosity 25) or UMAP (15 neighbors, minimum distance 0.4, 200 Epochs), followed by PhenoGraph clustering based on Euclidean distance. When comparing two groups, a two-class unpaired approach using SAM (Significance Analysis of Microarrays) was performed, with a maximum of 100 permutations and a False Discovery Rate (FDR) cutoff of 0.1.

## Immunohistochemistry

For immunohistochemical analyses of the CNS, mice were transcardially perfused with cold PBS. For LCP-induced demyelination experiments, the mice were additionally perfused with 4% PFA/1x PBS. The brain, spinal cord and retinotectal system were dissected and processed for immunofluorescence labelling. The tissue was post-fixed in 4% PFA/1x PBS at 4 °C for 24 h. After post-fixation, the tissue was dehydrated at 4 °C in 30% sucrose in PBS overnight. By means of liquid nitrogen-cooled 2-methylbutane, the tissue was frozen in tissue-Tek embedding medium and kept at -80 °C for storage. 10 μm (spinal cord, optic nerve), 12 μm (brain) thick cross cryostat sections (Leica) were obtained on glass slides and stored at -20 °C until further analyses. For immunohistochemical analyses of TGFα, Iba1 and GFAP, spinal cord cross sections were incubated in acetone for 10 min at -20 °C for post-fixation. After one washing step, the tissue was incubated in blocking buffer (5% BSA / 10% donkey serum / 0.3% Triton-X/1x PBS) for 1 h. Slides were incubated overnight at 4 °C with mouse anti- TGFα (1:100; Santa Cruz; sc36-134A), rat anti-GFAP (1:1000; Thermo Fischer

Scientific; #2.2B10), rabbit anti-Iba1 (1:1000; Abcam; #ab178846), rabbit anti-NeuN (1:500; Abcam; #ab177487) diluted in 1% BSA / 1% donkey serum / 0.3% Triton-X / 1x PBS. On the following day, three washing steps of 5 min each preceded the incubation with the secondary antibodies for 1 h: 1donkey anti-rabbit IgG AF488 (1:500; Thermo Fisher Scientific; #A21206), donkey anti-mouse IgG AF647 (1:500; Dianova; #715-605-151), goat anti-rat IgG Cy3 (1:500; Thermo Fisher Scientific; #A10522). During the further procedure, sections were washed three times for 5 min and then incubated with DAPI (Sigma, #D8417) diluted 1:100.000 in antibody solution for 10 min at RT and washed 3x in 1x PBS for 5 min. Finally, the slides were cover-slipped with Prolong Gold anti-fade (Thermo Fisher Scientific, #P36930) and stored at 4 °C for further analysis.

For analysis of SMI32, Olig2, or NeuN, spinal cord or optic nerve sections were incubated in acetone for 10 min at -20 °C for post-fixation. After washing the slides in 1x PBS for 5 min, they were incubated in blocking buffer (5% BSA / 10% donkey serum / 0.3% Triton-X/1x PBS) for 30 min. Slides were incubated overnight at 4 °C with mouse anti-SMI32 (1:1000; BioLegend; #801701), rabbit anti-Olig2 (1:200; Abcam; #ab109186) or rabbit anti-NeuN (1:500; Abcam; #ab177487) diluted in 1% BSA / 1% donkey serum / 0.3% Triton-X / 1x PBS. On the following day, three washing steps of 5 min each preceded the incubation with the secondary antibodies for 1 h donkey anti-mouse IgG AF647 (1:500; Dianova; #715-605-151), goat anti-rabbit IgG Cy3 (1:500; Thermo Fisher Scientific; #A10520), donkey anti-rabbit IgG AF647 (1:500; Dianova; #711-605-152), goat anti-rat IgG Cy3 (1:500; Thermo Fisher Scientific; #A10522). During the further procedure, sections were washed three times for 5 min. After this process, the cross sections were incubated with DAPI diluted 1:100.000 in antibody solution for 10 min at RT and washed 5x in 1x PBS for 3 min. Finally, the slides were cover-slipped with Prolong Gold anti-fade and stored at 4 °C for further analysis.

For Fluoromyelin staining, the Invitrogen™ FluoroMyelin™ Green Fluorescent Myelin Staining Kit (Thermo Fisher Scientific, #F34651) was used according to the manufacturers' instruction.

The retinae were dissected for immunofluorescence retinal flatmount analysis to quantify retinal ganglion cells. For this, the optic nerve was removed from the eye, which was further opened along the ciliary body. Cornea, vitreous body and lens were removed, and retinal pigment epithelium separated from retina. Retinal tissue was incised four times and placed in a 24-well plate filled with 1x PBS. After one washing step with 2% TritonX-100 in 1x PBS at room temperature, the retinae were frozen in fresh 2% TritonX-100 in 1x PBS at -80 °C for storage. For the visualization of retinal ganglion cells, the retinae were used for immuno-fluorescence free floating labelling against RBPMS. The tissue was thawed and washed two times in 1x PBS/ 0.5% Triton-X at room temperature with gentle agitation. The tissue was incubated in 500 µl blocking solution (1x PBS/ 5% bovine serum albumin (BSA)/ 10% donkey serum (DS)/ 2% Triton-X) for 1 h at room temperature. After incubation, the primary antibody was applied to the retinal tissue and incubated overnight at 4 °C with gentle agitation: rabbit anti-RBPMS (1:300; Merck, ABN1362) diluted in blocking buffer. After overnight incubation, the retinae were washed with 1x PBS/ 2% Triton-X for 5 min and then three times for 10 min with 1x PBS/ 0.5% Triton-X with gentle agitation. Secondary antibody was added to the tissue at a concentration of 1:500 and then incubated overnight at 4 °C on slow rocker: goat anti-rabbit IgG Cy3 (Thermo Fischer Scientific, #A10520). On the third day of staining, the retinae were washed three times for 30 min in 1x PBS at room temperature and subsequently transferred to microscope slides ensuring that the ganglion cell layer was facing up. Finally, the slides were cover-slipped with Prolong Gold anti-fade (Thermo Fisher Scientific, #P36930) and stored at 4 °C for further analysis.

## Image acquisition and analysis

Images of immunofluorescent labelled sections were acquired using the software Zen 3.0 (blue edition). Stainings were examined using a fluorescence microscope (Axio Observer Z1, Zeiss) at 5x, 10x, or 20x magnification. If not stated otherwise, three areas of 3-7 lumbar (L1 to L6) spinal cord cross sections per animal were investigated: anterior horn (area A) containing motor neurons, lateral horn with mainly white matter (area B), and gray matter of dorsal horn (area C) serving as a sensory processing region. Four images per region were taken. Overview images of whole cross spinal cord sections were acquired using a 40× objective and an Axio Zeiss imager M2. Cells were quantified manually in a blinded, unbiased manner by the same investigator using ImageJ software. Image processing was performed using Photoshop CS6 (Adobe).

## Magnetic resonance imaging of mouse spinal cords

All measurements were conducted, as previously published[26], using a preclinical 7 T MRI scanner (ClinScan 70/30, Bruker, Ettlingen, Germany) equipped with a dedicated mouse brain coil. During in vivo imaging, the animals were anesthetized with 3% isoflurane and maintained at 1.5% isoflurane throughout the entire imaging procedure. Respiratory rate was continuously monitored using breath sensors and maintained at a constant level. To stabilize body temperature, a heating circulator bath from Thermo Fisher Scientific (Waltham, USA) was used. An axial T1-weighted spin echo (SE) sequence was acquired twice before the administration of the contrast agent Gadolinium. The sequence parameters were as follows: field-of-view (FOV) = $35 \times 26.2 \times 14$ mm$^3$, voxel size = $0.078 \times 0.078 \times 0.7$ mm$^3$, matrix size = $448 \times 336 \times 20$, echo time (TE) = 9 ms, repetition time (TR) = 460 ms, bandwidth (BW) = 205 Hz/px, acquisition time (TA) = 2 min and 39 s. After the administration of the contrast agent, an axial T2-weighted turbo spin echo sequence was acquired with a FOV of $30 \times 30 \times 14$ mm$^3$, voxel size = $0.094 \times 0.094 \times 0.7$ mm$^3$, matrix size of $320 \times 320 \times 20$, TE = 53 ms, TR = 3800 ms, BW = 130 Hz/px, TA = 4 min and 37 s. Subsequently, the T1-weighted sequence was repeated four times to measure the contrast agent washout. Difference maps of the T1-weighted images before and after the application of the contrast agent were calculated on the scanner. These maps were used for manual segmentation using the MITK workbench[73] to calculate the lesion volume. Contrast enhancing lesions measured in this study represents the total volume of lesions with active inflammation.

## Multiplex analysis of cerebrospinal fluid

The multiplex analysis of specific analytes (including TGFα) in the cerebrospinal fluid (CSF) of both MS patients was conducted in collaboration with Thermo Fisher Scientific, as previously published[26].

## Enzyme-linked immunosorbent assay

The analysis of human TGFα in the CSF of both MS patients and controls (non-inflammatory disease) involved using a commercial human TGFα ELISA Kit (Thermo Fisher Scientific, #EHTGFAX10) following the manufacturer's instructions. The CSF samples were obtained from the inhouse MS Biobank, and they were diluted 1:1 for the analysis. Table 1 and Supplementary Table 1 contains the patient characteristics.

The analysis of mouse TGFα in the whole brain of EAE mice was performed using a commercial mouse TGFα ELISA Kit (Aviva, # OKCD06009) following the manufacturer's instructions.

For the analysis of mouse TGFα in the whole brain of EAE mice, mice were euthanized and perfused with cold 1× PBS. Brains were then isolated and mechanically diced using sterile razors in 500 µl PBS / 100 mg brains. The homogenised tissue was frozen at -20 °C for 24 h followed by two freeze-thaw cycles. The homogenised brain was centrifuged at 600 g for 5 min at 4 °C. The resulting supernatant was collected to detect TGFα protein concentration using a commercial

**Table 1 | Characteristics of MS patients and controls**

| Multiplex Analyses | | | | | |
|---|---|---|---|---|---|
| **Cohorts** | **Females** | **Age** | **Disease duration** | **EDSS** | **Treatment** |
| RRMS (47) | 37 (64.6%) | 33.1 [28.0: 38.0] | 2.4 [0.0: 3.0] | 1.0 [0.0: 3.0] | none |
| **ELISA** | | | | | |
| **Cohorts** | **Females** | **Age** | **Disease duration** | **EDSS** | **Treatment** |
| Controls (17) | 13 (76.4%) | 40.8 [30.0: 44.4] | none | none | none |
| RRMS (10) | 8 (80.0%) | 33.0 [23.5: 46.75] | 2.6 [0.0: 2.75] | 2.0 [0.0: 2.1] | none |

Characteristics of MS Patients "Females" indicates the absolute number and percentage of females in the group. "Age", "Disease duration" and "EDSS" are shown as mean, with 25% and 75% percentiles indicated in square brackets. "Treatment" indicates the absolute number and percentage of treated patients. No additional relevant comorbidities or pharmaceutical treatments were reported in patients or controls. Top: ELISA, Bottom: Multiplex analyses.

mouse TGFα ELISA Kit (Aviva, # OKCD06009) following the manufacturer's instructions.

### RNA isolation and RT-qPCR
Cells were lysed in 350 μl RLT buffer (Qiagen), and RNA was extracted using the RNeasy Mini Kit (Qiagen, #74004) as per the manufacturer's instructions. For cDNA synthesis, 500 ng of RNA from each sample was used with the High-Capacity cDNA Reverse Transcription Kit (Life Technologies, #4368813). Gene expression analysis was carried out using qPCR with the TaqMan Fast Advanced Master Mix (Life Technologies, #4444556). The following TaqMan-probes were used for target genes: *Gapdh* (Mm99999915_g1), *Actb* (Mm02619580_g1), *Tgfa* (Mm00446234_m1), *Tnf* (Mm00443258_m1), *Nos2* (Mm00440502_m1), *Il1b* (Mm00434228_m1), *Egfr* (Mm01187858_m1), *Lif* (Mm00434762_g1), *Ccl2* (Mm00441242_m1), *Cd86* (Mm00444540_m1), *Il6* (Mm00446190_m1), *Bcl2* (Mm00477631_m1), *Cxcl10* (Mm00445235_m1), *Csf2* (Mm01290062_m1), *Mbp* (Mm01266402_m1). qPCR data were analysed by the ΔΔCt method.

### Analysis of scRNA-seq data
To examine *Tgfa* gene expression across different CNS cell types, we analyzed publicly available single-cell RNA sequencing data from EAE mice, as published by Wheeler et al.[19]. The scRNA-seq data were assessed using Seurat[74], which included log-normalization and filtering of scRNA-seq profiles. After dimensionality reduction, unsupervised cell type clustering was performed using the Louvain algorithm. UMAP plots were generated using ggplot2 for visualization.

### Evaluation of results
Images of immunofluorescent labelled sections/ retinae were acquired using the software Zen 3.0 (blue edition). Stainings were examined using a fluorescence microscope (Axio Observer Z1, Zeiss) at 5x, 10x, or 20x magnification. Cells were quantified manually in a blinded, unbiased manner by the same investigator. The number of quantified cells was related to the area of measured sample using the cell counter plugin in ImageJ software (Rasband, W.S, ImageJ, National Institutes of Health, Bethesda, Maryland, USA). Image processing was performed using Photoshop CS6 (Adobe). The OMIQ platform was used for flow cytometric data analysis. Statistical examinations were carried out using GraphPad Prism 9.5.1.

### Statistics
For analysis of two groups, unpaired *t*-tests were applied. Multiple groups were analysed with One-way ANOVA with Tukey's or Dunnett's multiple comparisons test. For multiple testing, a Two-way ANOVA with Tukey's or Sidak's multiple comparisons test was applied. For EAE experiments a non-parametric Wilcoxin matched-pairs signed-rank test was used. Family-wise significance and confidence level was set at $p < 0.05$. If not stated otherwise, statistical examinations and linear regression analyses were carried out using GraphPad Prism 9 (v. 9.5.1).

Additional information on the study design, the number of replicates, and the statistical tests used are provided in the figure legends.

### Ethical approval
Experiments on human tissue were performed in accordance with the Declaration of Helsinki. Written informed consent was obtained from all participants. Approval was granted by the Ethics Committee of the University Hospital Erlangen (approval numbers: 20-484_1-Bio and 23-180-Bp). An overview of the sample characteristics is provided in Table 1 and Supplementary Table 1. The animal experiments were reviewed and approved by Bavarian State Authorities (approval numbers: 55.2.2-2532-2-1722; 55.2.2-2532-2-1927; 55.2.2-2532-2-1306; 55.2.2-2532.Vet_02-19-49).

### Reporting summary
Further information on research design is available in the Nature Portfolio Reporting Summary linked to this article.

## Data availability
All data supporting the findings of this study are available within the article and its Supplementary Information files. The processed human data are provided in the Supplementary Table submitted with this manuscript. The single-cell RNA-sequencing data used in this study are publicly available in the Gene Expression Omnibus (GEO) under the accession number GSE130119. Source data are provided with this paper.

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

## Acknowledgements

We thank all contributors and collaborators for their support of this study. We also acknowledge the Biobank of the Department of Neurology, as part of the Joint Biobank Munich and the German Biobank Node, for supporting the study. LL was funded by transregional research center provided by the German Research Foundation (DFG, German Research Foundation - Project ID 408885537 - TRR 274). ML and VeRo were funded by an ERC Starting Grant by the European research Council (HICI 851693). VeRo was supported by a Heisenberg fellowship and Sachmittel support provided by the German Research Foundation (Deutsche Forschungsgemeinschaft, DFG, RO4866-3/1, RO4866-4/1; -1/1, 2/-1, 3/1, 4/1, 5/1, 6/1 Project ID 401772351) as well as transregional and collaborative research centers provided by the German Research Foundation (DFG, Project ID 408885537-TRR274, Project ID 261193037-CRC1181, Project ID 270949263-GRK2162, Project ID 405969122-FOR2886, Project ID 505539112-GB.com). TT was funded by the Kommission für Klinische Forschung (KKF), Klinikum rechts der Isar as well as by the clinician scientist program (IZKF Erlangen) of the medical faculty of the Friedrich-Alexander-Universität Erlangen-Nürnberg. AP and OV were funded by the German Research Foundation (DFG, German Research Foundation - Project ID 270949263 - GRK2162). AP was supported by the German Research Foundation (DFG, German Research Foundation - Project ID 405969122 - FOR2886). FJQ received funding from the National Institutes of Health, by the NMSS and the Progressive MS Alliance (NS087867, ES025530, ES032323, AI126880 and AI149699). The Biobank of the Department of Neurology as part of the Joint Biobank Munich in the framework of the German Biobank Node supported the study.

## Author contributions

LL, ML, FiZu, TT, OV, AP, FP, NS, EN, UJN, JZ and ViRi performed in vitro, ex vivo and in vivo experiments. LL and FiZu performed immunohistochemical analyses of mouse tissue. LM performed primary mouse oligodendrocyte cultures. LN, FBL and JH performed MRI imaging of mouse spinal cords. LL and VeRo designed the study and edited the manuscript. JW, TB, AL, FrZu and FQ provided unique reagents and/or discussed and interpreted findings. LL and VeRo wrote the manuscript with input from all authors.

## Funding

## Competing interests

All authors declare no competing interests.

## Additional information

[1]Department of Neurology, University Hospital Erlangen, Friedrich-Alexander University Erlangen Nuremberg, Erlangen, Germany. [2]Center for Biotechnology, Khalifa University of Science and Technology, Abu Dhabi, United Arab Emirates. [3]Institute of Radiology, University Hospital Erlangen, Friedrich-Alexander University Erlangen Nuremberg, Erlangen, Germany. [4]Department of Diagnostic and Interventional Radiology, University Medical Center of the Johannes Gutenberg University Mainz, Mainz, Germany. [5]Department of Molecular Neurology, University Hospital Erlangen, Friedrich-Alexander University Erlangen Nuremberg, Erlangen, Germany. [6]Institute for Stroke and Dementia Research (ISD), University Hospital, LMU Munich, Munich, Germany. [7]Munich Cluster for Systems Neurology (SyNergy), Munich, Germany. [8]Ann Romney Center for Neurologic Diseases, Brigham and Women's Hospital, Harvard Medical School, Boston, MA, USA. [9]The Broad Institute of Harvard and MIT, Cambridge, MA, USA. ✉e-mail: Veit.rothhammer@fau.de

