## [Transparent Peer Review file · Nature Communications]

TGF α controls checkpoints in CNS resident and infiltrating immune cells to promote resolution of inflammation

Corresponding Author: Professor Veit Rothhammer

Version 0:

Reviewer comments:

Reviewer #1

(Remarks to the Author)

The authors present a study that proposes TGF α to be a critical factor leading to resolution of neuroinflammation using MOG-EAE and focal demyelination models. Results from a variety of in vitro and in vivo experimentations is presented, including immunohistochemistry, imaging, knockout mouse strains and both cellular and molecular analyses.

There is some interesting data presented, but as it stands, I find the manuscript extremely difficult to understand.

That major concern is the immunostaining presented in Figure 1e, which is very problematic. According to these images, there is no (or extremely little) overlap between TGF α staining and either Iba1+ or GFAP+ cells. So the question arises, what other unidentified cell type is expressing the large majority of the TGF α . Neurons? This observation – that microglia produce TGF α , is the heart of the study, and as this conclusion is questionable the whole study falls. This is not addressed in the flow cytometry data either, which only includes the immune+astrocyte compartment, but not other cell types.

Moreover, the microglia-specificity is based on CD11b-mediated knockout of TGF α using CRISPR, but the validation presented in Sup Fig 1 is unconvincing, and it is therefore extremely difficult to assess the specificity or efficiency of the knockout. Why was this not demonstrated using qPCR on sorted microglia or immunostaining?

It is difficult to reconcile the peak TGF α production recorded in both brain and spinal cord (Fig 1B and C) with a disease resolution and not pathogenesis. If TGF α was critical for the resolution it would be expected to be recorded during the recovery or late recovery stages instead.

In Fig.1J the proposed overlap of TGF α and Ki67 staining is not exact – different subpopulations appear to be represented in the different images. It is not possible from this image to understand the relevance of samples being from different timepoints.

Fig.1K right panel (PDGFR α /O4 oligo data) is not described in the Results.

Fig.3M It is not convincing with the images included of immunostaining of oligodendrocytes that there is a difference between IFN γ or IFN γ +TGF α treatments, despite the quantitation.

Fig.4C. The quantitation of the lesion volume does not seem to reflect the MRI images, at least at this resolution of images. Demarcation of the lesion in the MRI images (using a dashed line) might assist the reader in understanding if there is a difference.

Fig.4J Some of the most interesting and convincing data, the reduction in numbers of neutrophils and proliferating monocytes is barely discussed, and this warranted. .

Fig.4N There are 2 fluorescent colours in the image and only one cell type described (NeuN+ neurons) in the Figure legend.

Figure 4O. The human CSF data would seem to contradict the mouse EAE data. There is highest TGF α at peak of EAE disease, but reduced TGF α in the CSF of relapsing MS patients. This section of the text (line.238) states “we aimed to estimate the potential benefit of exogenous TGF α supplementation in patients suffering from MS”. According to Table 1 none

of the patients received any treatment, so this statement is incorrect as there was no augmentation of TGF α . It would have been more interesting to assess whether increased TGF α was associated with REMISSION in these patients. I would suggest removing all this human data.

There are numerous reports in the literature of the importance of TGF β in resolution of MOG-EAE. Was TGF β measured at all, as both TGF α and TGF β act together in other immunological settings (e.g. tumor biology, eosinophilia). At least this could be mentioned in the Discussion.

To link the proposed role of TGF α in reducing demyelination (Fig 2K and Fig 3P), why was fluoromyelin staining of the treated MOG-EAE mice in Fig.2b not included?

There are usually 5 mice used per group but there is no mention of how many times the experiments have been repeated. Are these results from single experiments throughout?

Reviewer #2

(Remarks to the Author)

This is a compelling study from a leading neuroimmunology group on the pleiotropic protective effects of TGF α in experimental autoimmune encephalomyelitis (EAE) and its translational implications for multiple sclerosis (MS). The experimental approach is innovative and informative. The significance of the work is appropriate for the journal.

However, major concerns regarding the details and validation of the experimental design need to be addressed, as outlined below. Additionally, the scope of the presented results as related to MS progression is overstated and would benefit from either an expansion of the dataset or qualification of the conclusions, as discussed in more detail below. The data, as currently presented, are more relevant to relapsing-remitting disease and relapse recovery, rather than MS progression.

To invoke the latter, ongoing TGF α mediated crosstalk in the extremely late recovery period (or chronic stable time point) would need to be implicated, and is not explored here. The return to baseline levels of microglial and astrocytic TGF α levels at this time point (Figs. 1G-H) in wild-type mice suggest against a suppressive role for TGF α in chronic progression but this could be tested by tracking the longitudinal course of EAE to the extremely late recovery period in WTs vs. microglial TGF α knock-outs.

Comments:

1) The reference to and repeated use of the term “CNS-compartmentalized inflammation” is misleading as the term is typically used in the clinical setting to describe a hypothesis that ongoing chronic inflammation from resident CNS glial cells (including microglia and astrocytes) and residual immune cells drives progressive disease behind a closed blood brain barrier. The EAE model and time points studied here involve an acute relapsing peak and recovery followed by a chronic stable course and therefore hold more relevance for relapse and recovery in relapsing-remitting disease. This is still very much an important translational focus for study, as discussed more below. However, as such, references to TGF α as a pathway with translational potential against MS progression (line 267-270, line 308-309, line 341-342) feel over-stated and not supported by the design of the study.

2) The Methods should include more details: including blinding, validation of knock-outs and cell specific cultures, and anatomical aspects of IHC analysis, as outlined below.

a. Details on blinding to genotype/intervention in EAE and flow cytometry are necessary – while mentioned for IHC analysis, they are not included for these other experiments. EAE scoring in particular is susceptible to unconscious bias if group identity is known. If blinding was not possible or could not be maintained for a given experiment, reasons should be provided.

b. EAE experiments typically have high variability. A literature standard is to repeat the experiment 2-3 times. The EAE experiments in Figs. 2 and 4 describe only one experiment per comparison with a fairly low n (we typically find n=8 per group necessary to achieve reproducible results). A given experiment with the appropriate n should be replicated and also representative of at least 2-3 experiments.

c. Statistical analysis of EAE should be specified in Figs. 2 and 4 legends or in the Methods section. A non-parametric Wilcoxin matched-pairs signed-rank test is typically used.

d. Validation of microglial TGF α and astrocytic Erbb1 knock-down in lentiviral experiments has not been included here. If validated in prior references, this should be noted; if not, validation could be performed using IHC, Western blotting or flow cytometry.

e. Scramble sgRNA sequences should be included in the Methods in addition to experimental Tgfa and Erbb1 sequences (Line 399).

f. Methods to validate purity of microglial, astrocyte and oligodendrocyte cultures are not mentioned, nor are images (ie.

antibody staining for cell-specific markers) provided. These should be included here or referenced, if performed previously by the group.

g. Analysis of spinal cord IHC in EAE should include the number of sections analyzed per animal (n) and anatomical localization of the regions of analysis (ie. whether cervical, thoracic, or lumbar; grey or white matter; which tracts measured). EAE IHC images in Figs. 2H-I and Figs. 4L-N should include lower power views (eg. 40X) to show recognizable architectural features of the spinal cord (gray vs. white matter, dorsal vs. ventral landmarks) to demonstrate where the regions of analysis are taken from.

3) The decision to analyze timepoints from day post-injection rather than day post-disease onset (which removes inter-animal variability in disease onset) should be justified. An analysis using day post-disease onset might be informative to demonstrate that the time course of TGF α mediated effects appears similar using both approaches.

4) In Fig. 1, the extremely late recovery time-point should be specified (day post-injection). For clarity, all time points in Fig. 1 would be ideally described in the Methods section for EAE and not just the Fig. 1A image.

5) How dosing for intranasal TGF α was determined should be described in more detail. Was it derived from prior mouse studies with validated CNS levels and dose titrations? If not, this information should be included.

6) In Fig. 2, the time point of flow and IHC experiments demonstrating differences in proinflammatory cell count and demyelination/repair is not specified in the text or the Fig. legend (presuming this is the recovery time point only?)

7) EAE in the lentivirally-induced microglial TGF α knock-outs shows a protective effect for TGF α in recovery, but not peak, whereas intranasal TGF α shows protective effects in both peak and recovery. Is this due to a specific role for microglial TGF α in recovery? Or, could this difference be due to a delayed onset of TGF α knock-down using the lentiviral technique? How might the benefits to peak disease from intranasal TGF α be explained – could it be that it is a supraphysiologic dose? How specific are TGF α 's protective effects to the peak and recovery time points (ie. when do they start to occur)?

8) Experiments in Fig. 4 measure changes in inflammatory cell counts and demyelination/repair at the recovery time point for TGF α i.n. animals vs. controls. Given the difference in scores seen at peak, it may be informative to measure these changes at the peak time point to characterize when and to what extent these protective effects occur. Additionally, in comparing the analysis to Fig. 2, IL-17 measurements should be included in Fig. 4D. (IL-10 would also be interesting if available, as this would show potential changes in regulatory cells). Fluoromyelin should also be measured in experiments of Fig. 4, as was done in Fig. 2K, while NeuN (as measured in Fig. 4N) should be measured in Fig. 2.

9) For MRI data in Fig. 4C, use of contrast enhancement to measure lesion volume should be qualified and noted to reflect the volume of active inflammation at this recovery time point, not necessarily the total lesion volume (line 207-209).

10) The MS CSF data should specify when samples were taken (during relapse, acute recovery or between attacks) as this has implications for relative TGF α levels contributing to the severity of relapse vs. ongoing recovery/repair.

11) MS subjects do not seem ideally age-matched as the average ages of both MS cohorts is younger than controls. Details on how the data was or could be age- and gender-matched would be helpful.

12) The Discussion would benefit from a more detailed discussion of downstream effectors of TGF α -EGF signaling and how this pathway might have parallel effects 1) steering proinflammatory CD4+ T cell differentiation to IL-17-, IFN γ - and GM-CSF-producing subtypes and 2) driving oligodendrocyte differentiation and repair.

13) In the Discussion, the limited "availability of...treatments...designed to promote resolution" seems to represent the strongest translational application of this work and can be highlighted by the fact that steroids remain the only clinical option for this, citing data that steroids speed up the time course of clinical recovery during MS relapse but do not change the ultimate disability outcome.

14) The statement in the Discussion that CNS penetrating TGF α "preserves neurons and oligodendrocytes and reduces lesion volume in the spinal cord, consequently improving clinical outcome" holds for the acute course of EAE peak and recovery (day 25 post-immunization) in this study, but does not exactly demonstrate its benefit in "chronic neuroinflammation," as noted in line 74. This statement would be more accurate and carry stronger translational promise if EAE differences with TGF α i.n. were shown to persist to the extremely late recovery time-point. A comparison between high dose steroids and TGF α might even reveal useful pre-clinical information regarding the relative magnitude and duration of these respective treatments.

Reviewer #3

(Remarks to the Author)

The authors have provided evidence that microglia are the primary source of TGF in the murine CNS, its production is increased on acute inflammation and reduces with inflammatory resolution, and that TGF is protective in EAE. Microglia with TGF -KO in EAE impaired recovery. They detected an increase of infiltrating cells (CD4+ T-cells (with elevated IL-17 and INF), macrophages, pro-inflammatory monocytes, pro-inflammatory B cells and neutrophils in TGF - mice. They also

measured a decrease of oligodendrocyte lineage cells including olig2+ oligodendrocytes. They provide further evidence for the protective properties of TGF β by utilising lysolecithin in vitro and in vivo models of demyelination, showing that addition of TGF β reduced demyelination in comparison to vehicle treated controls. Their results suggest that microglial TGF β has a role in recovery after inflammatory damage in EAE, possibly through pathways which regulate pro-inflammatory T cells/ myeloid cells. Additionally, they provide evidence for a neuroprotective role for TGF β as oligodendrocyte lineage cell and neuronal cell death was lessened with TGF β addition, consequently, affecting de/remyelination.

They also demonstrate that TGF β can be administered intranasally and that via this delivery method it can penetrate the CNS. Its addition promotes glial-mediated pathways which contribute to neuron survival and clinical recovery in EAE.

Importantly, this work may be translatable to human MS as they demonstrate that TGF β is reduced in the CSF of pwRRMS. The results, methodology and conclusion provided are clear, but a few points need addressing:

1. General – is microglia being used as a catch-all term for microglia and CNS macrophage-like cells? Needs clarification when term first used if so.

2. There is a typo in the sentence over lines 492 – 494, this typo makes this portion of the methodology unclear.

3. Figure 1 – include which regions of brain/ level of spinal cord were used for analysis of TGF β expression and how many sections were used for the immunohistochemical analysis.

4. Include abbreviations & meanings in figure legends, e.g., figure 3 – define RBPMS.

5. Discussion – lines 307 – 309 “This highlights the pleiotropic effects of TGF α and provides insights into its potential to mitigate disease progression”. This statement is overreaching based on the evidence presented in the paper. The authors include samples from pwRRMS and inflammatory/ toxin-induced demyelination models which do not accurately capture/model the progressive phase of human MS. There is evidence in the literature to show that current DMTs, that reduce annualised relapse rate, do not prevent disease progression. Therefore, this conclusion needs to be reworded to reflect that this is not something that can be assumed and instead provides evidence to warrant further investigation in the context of progressive disease.

This work lacks some translatability as the authors have not provided evidence of TGF β expression in human MS or control CNS tissue. Although this limitation should be addressed, it is outside of the scope of the present study and its exclusion is not sufficient to prevent publication.

It is my view, that with the minor alterations listed above, that this work is suitable for publication.

Version 1:

Reviewer comments:

Reviewer #1

(Remarks to the Author)

The manuscript is greatly improved and with all the additional data and discussion I am satisfied that the current version is worthy of publication.

Reviewer #2

(Remarks to the Author)

The authors have performed the appropriate additional experiments and revisions to make this study suitable for publication. Concerns and comments from my initial review have been thoroughly addressed.

I agree with including Figure 1R in the revised manuscript as it demonstrates that timing of TGF α knock down before and during EAE induced upregulation of TGF α is sufficient to impair protective pathways. These experiments support the later findings that TGF α treatment pre- and post-induction targets protective pathways accessible at both timepoints.

Some additional minor suggestions:

In referencing data in 4Q, I would recommend changing "disease severity" to "disability at the time of relapse". "Disease severity" may cause confusion for clinician readers who typically associate the term with a longitudinal level of accumulated disability (between relapses and/or during progression).

Consider including that a limitation of the study is that the effects of TNF α on CD8+ T cells, specifically regulatory CD122+ cells, were not explored. As this population has been implicated in controlling the remission phase of EAE (<https://pubmed.ncbi.nlm.nih.gov/31391585/>), this question would be an interesting effector pathway to explore as a future direction.

Reviewer #3

(Remarks to the Author)

I don't have further comments

Reviewer #1

The authors present a study that proposes TGF α to be a critical factor leading to resolution of neuroinflammation using MOG-EAE and focal demyelination models. Results from a variety of in vitro and in vivo experimentations is presented, including immunohistochemistry, imaging, knockout mouse strains and both cellular and molecular analyses.

There is some interesting data presented, but as it stands, I find the manuscript extremely difficult to understand.

That major concern is the immunostaining presented in Figure 1e, which is very problematic. According to these images, there is no (or extremely little) overlap between TGF α staining and either Iba1+ or GFAP+ cells. So the question arises, what other unidentified cell type is expressing the large majority of the TGF α . Neurons? This observation – that microglia produce TGF α , is the heart of the study, and as this conclusion is questionable the whole study falls. This is not addressed in the flow cytometry data either, which only includes the immune+astrocyte compartment, but not other cell types.

We thank the reviewer for the thorough review and the constructive feedback on our manuscript. We appreciate the comments and opportunity to address the concerns brought forward.

We acknowledge that Figure 1e has been poorly chosen and apologize for any confusion this may have caused, suggesting microglia as minor producers of TGF α only. To clarify this important point, we have performed additional analyses and experiments to define the main cellular producers of TGF α in the CNS. First, we performed analyses on a published dataset of single cell sequencing during peak of experimental autoimmune encephalomyelitis (Wheeler et. al., 2020) focusing on Tgf α expression within the CNS. Indeed, Tgf α expression was mostly expressed in microglia (49.8%) compared to other TGF α -producing cell types, such as oligodendrocytes (10.34%), endothelial cells (21.7%), astrocytes (4.84%) neutrophils (3.78%) and neurons (5.76%), corroborating microglia as key producers of TGF α in the CNS (new Fig. 1j). Next, we conducted additional immunohistochemical stainings for microglia, astrocytes, and neurons, further characterizing the proportional contribution of each cell type of TGF α production during different stages of the disease. These analyses revealed cell-specific dynamics of TGF α , showing that even though neurons and astrocytes do contribute to TGF α production during EAE, microglia remain its predominant source, particularly during peak and remission stages (Fig. 1g-i) in the revised manuscript). Thus, in these transcriptomic and protein analyses, microglia are the main producers of TGF α in the CNS at the peak of EAE. Indeed, validating our initial results, while also including novel data on additional sources of TGF α in the CNS. Given these analyses, we have now chosen more representative images to show overlap of microglial Iba1 with TGF α at different disease stages, which are now presented in Fig. 1e and quantified in Fig. 1f-i.

Finally, we would like to kindly disagree with the statement that “microglia produce TGF α , is the heart of the study”. Conversely, we would like to suggest that, even though microglia do produce TGF α to a great extent and are indeed the major cellular source of TGF α in the CNS during autoimmune inflammation, the major point of our study is that TGF α - be it microglia-derived or of other cellular sources - controls several responder populations relevant to tissue recovery and is thus of therapeutic potential for the resolution of autoimmune CNS inflammation. We have added these datasets and notions in the revised version of the manuscript throughout the result part as well within the revised version of the discussion.

Moreover, the microglia-specificity is based on CD11b-mediated knockout of TGF α using CRISPR, but the validation presented in Sup Fig 1is unconvincing, and it is therefore extremely difficult to assess the specificity or efficiency of the knockout. Why was this not demonstrated using qPCR on sorted microglia or immunostaining?

We thank the reviewer for bringing up this important point. To address these concerns of knock-out specificity, we have supplemented our flow cytometric analyses with immunohistochemistry as well as

qPCR analyses on sorted microglia as suggested by the reviewer. Indeed, both show significant reduction of TGF α on RNA and protein level, verifying the validity of the CRISPR-based knock-out approach (Suppl. Figs. 2a-d).

It is difficult to reconcile the peak TGF α production recorded in both brain and spinal cord (Fig 1B and C) with a disease resolution and not pathogenesis. If TGF α was critical for the resolution it would be expected to be recorded during the recovery or late recovery stages instead.

We thank the reviewer for given us the opportunity to clarify this point. Our data show that knock-down of microglial TGF α worsens disease. Thus, deficiency/reduction of TGF α is not part of disease pathogenesis, but participates in promoting its resolution instead. In these lines, our *in vitro* studies also show anti-inflammatory and tissue-protective effects of TGF α on different cellular subsets (Fig. 3). Finally, intranasal application of TGF α leads to an ameliorated course of the disease.

Finally, in order to determine disease stage-specific effects of TGF α , we have performed additional knockout experiments of TGF α at the peak of EAE (Figure 1R). Indeed, as seen with the experiment knocking out TGF α before disease onset, TGF α knock-out at peak of disease also lead to a failure to recover.

Figure 1R. Absence of microglia-derived TGF α impacts recovery from acute CNS inflammation. *left*, delivery of lentiviruses (CD11b::scrambl; CD11b::TGF α) via intracerebroventricular (i.c.v.) injection before EAE induction (manuscript Fig. 2b) and EAE development in mice transduced with CD11b::scrambl (n=5) or CD11b::TGF α (n=5). *right*, delivery of lentiviruses via i.c.v. injection at peak of EAE and disease development in mice transduced with CD11b::scrambl (n=5) or CD11b::TGF α (n=5). Data shown as mean \pm SEM. ***P < 0.001; ****P < 0.0001.

Together, these experiments validate anti-inflammatory and tissue-protective effect of TGF α even at different stages of the disease, irrespective of its expression stage and dynamics. If the editor and/or the reviewer support conceptual relevance of this dataset, we are happy to include it in the revised version of the manuscript.

In Fig.1J the proposed overlap of TGF α and Ki67 staining is not exact – different subpopulations appear to be represented in the different images. It is not possible from this image to understand the relevance of samples being from different timepoints.

We agree with the reviewer that the overlap between TGF α and Ki67 staining is only partial and not exact in this UMAP plot, in which microglia have been subsampled. However, please note that these images do not represent different timepoints, as suggested by the reviewer, but the UMAP plots show expression levels of the markers analyzed (TGF α , CD45, Ki67) over the microglia population instead. Indeed, the plot shows that most TGF α positive microglia co-express the proliferation marker Ki67, and may thus be classified as “proliferative” microglia, which is also supported by the differential regulation

of CD45 over these subsets. This underlines the protective functions of TGF α -producing microglia associated with a proliferative phenotype, which has been found in previous studies in spinal cord injury focusing on the role of proliferating microglia as well (Bellver-Landete et al., Nat Commun. 2019 Jan 31;10(1):518). We have included this study in the revised version of the manuscript and adjusted the statement (**line 118-120**) accordingly.

Fig.1K right panel (PDGFR α /O4 oligo data) is not described in the Results.

We apologize for this oversight and have revised the manuscript accordingly (lines 125-127).

Fig.3M It is not convincing with the images included of immunostaining of oligodendrocytes that there is a difference between IFN γ or IFN γ +TGF α treatments, despite the quantitation.

We thank the reviewer for this important point. We have included new images to better visually support the quantitative analysis of this figure.

Fig.4C. The quantitation of the lesion volume does not seem to reflect the MRI images, at least at this resolution of images. Demarcation of the lesion in the MRI images (using a dashed line) might assist the reader in understanding if there is a difference.

We thank the reviewer for this suggestion and have adjusted the figure accordingly.

Fig.4J Some of the most interesting and convincing data, the reduction in numbers of neutrophils and proliferating monocytes is barely discussed, and this warranted.

We thank the reviewer for bringing this important point forward. We have expanded the discussion section to highlight the attenuated numbers of neutrophils and proliferating monocytes in response to intranasal TGF α treatment and have added a dedicated subsection in the discussion that focuses on the reduction of neutrophils and proliferating monocytes (lines 341 -346).

Fig.4N There are 2 fluorescent colours in the image and only one cell type described (NeuN+ neurons) in the Figure legend.

We thank the reviewer for the opportunity to clarify this and apologize for the inaccuracy in our description. In the revised version of our manuscript, we have included a detailed description of the immunostaining using DAPI (blue) for nuclear staining (Fig. 1e, Fig. 2l, Fig. 3m, Fig.4o, Suppl. Fig 1, 2, 4).

Figure 4O. The human CSF data would seem to contradict the mouse EAE data. There is highest TGF α at peak of EAE disease, but reduced TGF α in the CSF of relapsing MS patients. This section of the text (line.238) states “we aimed to estimate the potential benefit of exogenous TGF α supplementation in patients suffering from MS”. According to Table 1 none of the patients received any treatment, so this statement is incorrect as there was no augmentation of TGF α . It would have been more interesting to assess whether increased TGF α was associated with REMISSION in these patients. I would suggest removing all this human data.

We thank the Reviewer for the opportunity to clarify this point. In our animal studies, we explore the role of microglia-derived TGF α on per-cell level, which is elevated over the course of EAE as compared to

control conditions. In line with this, we have previously shown that TGF α is upregulated in human MS tissue, mirroring our results in EAE (Rothhammer et al., Nature 2018).

Here, we present novel data on TGF α levels in the CSF of treatment-naïve MS patients during relapse as compared to controls, with the goal to identify a potential relative deficit of TGF α in the CSF during acute inflammation, which could be supplemented by intranasal application. Indeed, TGF α levels were reduced during relapse, while the inverse correlation of CSF TGF α levels with disease severity suggested more severe disease alongside reduced TGF α levels. Together, these results argue in favor of a potential benefit of exogenous TGF α administration with CSF-targeted TGF α administration.

Please also note that CSF was collected in treatment-naïve MS patients during relapse due to the fact that most MS patients get lumbar punctures during their first relapse only, when MS diagnosis is made to rule out differential diagnoses of CNS inflammation such as infections. Since these patients are treatment-naïve by definition, we were not able to make a correlation of the effects of disease-modifying treatments on TGF α levels, as assumed by the reviewer: **“This section of the text (line.238) states “we aimed to estimate the potential benefit of exogenous TGF α supplementation in patients suffering from MS”. According to Table 1 none of the patients received any treatment, so this statement is incorrect as there was no augmentation of TGF α ”.** We thus would like to politely contradict the critique of the reviewer, since we did not aim at evaluating treatment-related augmentation of TGF α , as assumed by the reviewer, but a reduction of TGF α during relapse in the CSF instead, with the goal to determine whether supplementation of TGF α might be of potential relevance to ameliorate disease. For more clarity, we have adjusted this sentence and used the term “treatment-naïve” in the revised version of the manuscript.

Together, we do believe that the CSF data shown here as well as the correlation of TGF α levels with disease severity during relapse in a treatment-naïve cohort of MS patients are of relevance and highlight the potential use of TGF α as a mediator to promote lesion resolution using a CSF-accessible application mode such as intranasal application. We thus suggest to keep this dataset, if the editor agrees, as it opens novel avenues of translational research based on our study.

There are numerous reports in the literature of the importance of TGF β in resolution of MOG-EAE. Was TGF β measured at all, as both TGF α and TGF β act together in other immunological settings (e.g. tumor biology, eosinophilia). At least this could be mentioned in the Discussion.

We appreciate the reviewer bringing up the important role of TGF β in the context of MOG-EAE and have included a discussion on the potential involvement of TGF β in our manuscript (**lines 376-381**).

Additionally, we have performed single cell sequencing analyses of Tgfb expression during EAE to delineate different TGF β -producing cell populations at peak of disease (Fig. 2R). Indeed, these analyses show that in addition to microglia and macrophages, monocytes, neutrophils, endothelial cells, as well as T cells produce Tgfb within the inflamed CNS. This pattern suggests broad expression of Tgfb within the CNS, alongside its multiple functions described in other research (e. g. Chen B, Fron Immunol 2022; as well as other research cited in the revised version of the manuscript). Since this lies beyond the scope of the manuscript, we suggest to leave out this dataset on Tgfb expression in the CNS, but are happy to include it, should the reviewer or the editor wish us to do so.

Figure 2R. Tgfb expression in different CNS cells at peak of EAE. Analysis of scRNA-seq dataset (Wheeler et al., 2020) from CNS samples at peak of EAE, showing expression levels of Tgfb across distinct cell clusters: microglia/macrophages (MG/mac), monocytes/macrophages (mono/mac), neutrophils, endothelial cells, T cells, dendritic cells (DCs), neurons, astrocytes, oligodendrocytes, stromal cells, and smooth muscle cells.

To link the proposed role of TGF α in reducing demyelination (Fig 2K and Fig 3P), why was fluoromyelin staining of the treated MOG-EAE mice in Fig.2b not included?

We thank the reviewer for this constructive feedback and for highlighting this important point. In response, we have performed additional Fluoromyelin stainings to assess myelination in the spinal cord of TGF α -treated EAE mice. In the revised version of the manuscript, we have included representative images and quantification of the myelination status (Fig. 4n), depicting augmentation of myelinated tissue in TGF α treated mice.

There are usually 5 mice used per group but there is no mention of how many times the experiments have been repeated. Are these results from single experiments throughout?

We apologize for this oversight and have now included the number of repetitions of each experiment in the revised version of the manuscript (n=3 or more) (lines 426-428).

Reviewer #2

This is a compelling study from a leading neuroimmunology group on the pleiotropic protective effects of TGF α in experimental autoimmune encephalomyelitis (EAE) and its translational implications for multiple sclerosis (MS). The experimental approach is innovative and informative. The significance of the work is appropriate for the journal.

We thank the reviewer for his positive assessment of our manuscript.

However, major concerns regarding the details and validation of the experimental design need to be addressed, as outlined below. Additionally, the scope of the presented results as related to MS progression is overstated and would benefit from either an expansion of the dataset or qualification of the conclusions, as discussed in more detail below. The data, as currently presented, are more relevant to relapsing-remitting disease and relapse recovery, rather than MS progression.

We thank the reviewer for these comments and have addressed the concerns as outlined in detail below.

To invoke the latter, ongoing TGF α mediated crosstalk in the extremely late recovery period (or chronic stable time point) would need to be implicated, and is not explored here. The return to baseline levels of microglial and astrocytic TGF α levels at this time point (Figs. 1G-H) in wild-type mice suggest against a suppressive role for TGF α in chronic progression but this could be tested by tracking the longitudinal course of EAE to the extremely late recovery period in WTs vs. microglial TGF α knock-outs.

*We thank the reviewer for the detailed and constructive feedback. As suggested by the reviewer, we have performed an additional knock-out EAE experiment, which has been followed until the very late stage of recovery. Indeed, while TGF α levels were still reduced in the knock-out condition, the effects of TGF α deficiency were maintained until the very late stage of recovery. This novel experiment has been included in the revised version of the manuscript (**Suppl. 2g**), and argues in favor of a role of TGF α in chronic progression. To further corroborate this interpretation of our data, we have performed an additional EAE experiment knocking out TGF α after disease initiation. Indeed, also in this paradigm, we still observed a disease-worsening effect of TGF α deficiency during later stages (**Fig. 1R**), together suggesting a protective effect of TGF α even during late and extremely late stages of autoimmune inflammation.*

Comments:

1) The reference to and repeated use of the term “CNS-compartmentalized inflammation” is misleading as the term is typically used in the clinical setting to describe a hypothesis that ongoing chronic inflammation from resident CNS glial cells (including microglia and astrocytes) and residual immune cells drives progressive disease behind a closed blood brain barrier. The EAE model and time points studied here involve an acute relapsing peak and recovery followed by a chronic stable course and therefore hold more relevance for relapse and recovery in relapsing-remitting disease. This is still very much an important translational focus for study, as discussed more below. However, as such, references to TGF α as a pathway with translational potential against MS progression (line 267-270, line 308-309, line 341-342) feel over-stated and not supported by the design of the study.

We thank the reviewer for bringing up this important point and apologize for these overstatements. In response, we have removed references to “CNS-compartmentalized inflammation” and termed them as “CNS-intrinsic inflammation”. Moreover, we have tuned down the statements as to the translational potential of TGF α throughout the manuscript.

2) The Methods should include more details: including blinding, validation of knock-outs and cell specific cultures, and anatomical aspects of IHC analysis, as outlined below. *In response to the reviewer's comment, we have included specific descriptions of the experiments in the revised section of the methods. Additionally, we have added validation of knockouts in **Supplementary Fig. 2a, d, and h.***

a. Details on blinding to genotype/intervention in EAE and flow cytometry are necessary – while mentioned for IHC analysis, they are not included for these other experiments. EAE scoring in particular is susceptible to unconscious bias if group identity is known. If blinding was not possible or could not be maintained for a given experiment, reasons should be provided.

*We thank the reviewer for his comment. All scoring procedures were performed in a blinded manner. We have included this as well as specific descriptions on EAE scoring and flow cytometry in the Methods section of the revised manuscript (**Lines 421-423**).*

b. EAE experiments typically have high variability. A literature standard is to repeat the experiment 2-3 times. The EAE experiments in Figs. 2 and 4 describe only one experiment per comparison with a fairly low n (we typically find n=8 per group necessary to achieve reproducible results). A given experiment with the appropriate n should be replicated and also representative of at least 2-3 experiments.

*We thank the reviewer for his assessment and for highlighting the need to repeat EAE experiments due to the variability typically observed in these studies. In the revised manuscript, we have included information about the repetition of our EAE experiments (n=3 or more) and have increased the number of animals per experiment up to 5 (**Lines 425-428**). Due to legislative requirements, we were not allowed to increase group sizes over 5 animals per experiment, but were still able to produce meaningful and statistically significant results, as shown in the respective EAE experiments here.*

c. Statistical analysis of EAE should be specified in Figs. 2 and 4 legends or in the Methods section. A non-parametric Wilcoxin matched-pairs signed-rank test is typically used.

*We thank the reviewer for this valuable suggestion. We agree that using the Wilcoxon matched-pairs signed-rank test is an appropriate and commonly accepted method for analyzing EAE data. In the revised version of the manuscript, we have updated the statistical analysis and figure legends for **Figs. 2 and 4**, as well as the Methods section (**Line 879-880**), to clearly specify the statistical analyses performed.*

d. Validation of microglial TGF α and astrocytic Erbb1 knock-down in lentiviral experiments has not been included here. If validated in prior references, this should be noted; if not, validation could be performed using IHC, Western blotting or flow cytometry.

*In response to the reviewer's comment, we have now included validation of knock-outs on protein level. These datasets have been included in the revised version of the manuscript (**Lines 138, 139 and Suppl. Figs. 2a-d and 4i**).*

e. Scramble sgRNA sequences should be included in the Methods in addition to experimental TGF α and Erbb1 sequences (Line 399).

*We have added this information in the revised version of the manuscript (**Line 467**) and thank the reviewer for bringing up this point.*

f. Methods to validate purity of microglial, astrocyte and oligodendrocyte cultures are not mentioned, nor are images (ie. antibody staining for cell-specific markers) provided. These should be included here or referenced, if performed previously by the group.

We thank the reviewer for elaborating on this point. While we have previously published purity levels of our glial cultures (e. g. Rothhammer et al., Nat Med 2016, Rothhammer 2018, Linnerbauer 2024 etc), we have also included this information in the revised version of the manuscript (Suppl. Fig. 3a).

g. Analysis of spinal cord IHC in EAE should include the number of sections analyzed per animal (n) and anatomical localization of the regions of analysis (ie. whether cervical, thoracic, or lumbar; grey or white matter; which tracts measured). EAE IHC images in Figs. 2H-I and Figs. 4L-N should include lower power views (eg. 40X) to show recognizable architectural features of the spinal cord (gray vs. white matter, dorsal vs. ventral landmarks) to demonstrate where the regions of analysis are taken from.

We thank the reviewer for the suggestions and have included this information as well as lower magnification images in the revised version of the manuscript (Suppl. Figs. 2f, 4k).

3) The decision to analyze timepoints from day post-injection rather than day post-disease onset (which removes inter-animal variability in disease onset) should be justified. An analysis using day post-disease onset might be informative to demonstrate that the time course of TGF α mediated effects appears similar using both approaches.

*We thank the reviewer for this assessment. We have included an additional analysis using the day post-disease onset to demonstrate that the TGF α -mediated effects follow a similar time course with both approaches. We have added the graph to **Suppl. Fig. 2e**.*

4) In Fig. 1, the extremely late recovery time-point should be specified (day post-injection). For clarity, all time points in Fig. 1 would be ideally described in the Methods section for EAE and not just the Fig. 1A image.

We thank the reviewer for this comment and have included the specific timepoints in our methods section of the revised manuscript (Lines 422-423).

5) How dosing for intranasal TGF α was determined should be described in more detail. Was it derived from prior mouse studies with validated CNS levels and dose titrations? If not, this information should be included.

We thank the reviewer for the opportunity to clarify this important point. The intranasal treatment dosage was chosen based on our previous studies on intranasal application using similar-sized proteins (e.g. IFN β (Rothhammer et al., Nature Medicine 2016), or HB-EGF (Linnerbauer et al., Nature Communications 2024). We have included this information into the revised version of the manuscript as suggested by the reviewer (Lines 233-235).

6) In Fig. 2, the time point of flow and IHC experiments demonstrating differences in proinflammatory cell count and demyelination/repair is not specified in the text or the Fig. legend (presuming this is the recovery time point only?)

We thank the reviewer and apologize for our oversight. These analyses were performed during the recovery stages, as assumed correctly by the reviewer. We have included this information in both the text (Line 145) and the figure legend in the revised version of the manuscript.

7) EAE in the lentivirally-induced microglial TGF α knock-outs shows a protective effect for TGF α in recovery, but not peak, whereas intranasal TGF α shows protective effects in both peak and recovery. Is this due to a specific role for microglial TGF α in recovery? Or, could this difference be due to a delayed onset of TGF α knock-down using the lentiviral technique? How might the benefits to peak disease from intranasal TGF α be explained – could it be that it is a supraphysiologic dose? How specific are TGF α 's protective effects to the peak and recovery time points (ie. when do they start to occur)?

*We thank the reviewer for their insightful comment. Indeed, in our knock-out experiments, we see major effects of Tgf α knock-down during later stages of the disease, while peak EAE scores remain unaffected, arguing in favor of a pro-resolving effect of endogenous TGF α rather than the prevention of lesion formation. Together with our in vitro data on the effects of TGF α on oligodendrocytes and neurons in Fig. 3, the new experiment knocking-out Tgf α at peak stages of EAE (Fig. 1R) supports the interpretation that **endogenous TGF α** acts on the resolution of inflammation rather than by inhibiting lesion formation as reflected by peak EAE scores. By contrast, in our **exogenous application studies** using early intranasal application of TGF α intervention, we observed amelioration of peak disease scores as well as late stage effects. This may be explained by the fact that exogenous TGF α targets both actively infiltrating immune cells and CNS-resident cells, as suggested in numbers and cytokine production of infiltrating immune cells, which was more pronounced upon exogenous administration as compared to the reduction of infiltrating immune cells when reducing endogenous production. To follow up on the reviewer's suggestion of a potential supra-physiologic dosing of TGF α , we measured protein content of TGF α in intranasally treated mice, but were not able to detect significant differences (**Suppl Fig. 4c**), potentially due to the short half-life of TGF α , which might be comparable to TGF α -PE38 which is about 10-20 minutes¹. Thus, it is very well possible that exogenous administration leads to a supra-physiological TGF α concentration in the CNS temporarily. Since this is hard to address experimentally, we have added this potential interpretation in the revised version of the discussion. Finally, to experimentally address time-point specific effects of TGF α , we have performed pre-onset as well as post-onset intervention studies in Figs. 4b and 4h, respectively. Indeed, we observed clinically measurable differences between 5 (presymptomatic) and 3 (symptomatic) days after treatment onset, suggesting pro-resolving effects of exogenous TGF α administration. We have incorporated these novel datasets as well as interpretations in the revised version of the manuscript.*

8) Experiments in Fig. 4 measure changes in inflammatory cell counts and demyelination/repair at the recovery time point for TGF α i.n. animals vs. controls. Given the difference in scores seen at peak, it may be informative to measure these changes at the peak time point to characterize when and to what extent these protective effects occur. Additionally, in comparing the analysis to Fig. 2, IL-17 measurements should be included in Fig. 4D. (IL-10 would also be interesting if available, as this would show potential changes in regulatory cells). Fluoromyelin should also be measured in experiments of Fig. 4, as was done in Fig. 2K, while NeuN (as measured in Fig. 4N) should be measured in Fig. 2.

*We thank the reviewer for the comments and we have included the data as requested: we detected significant improved myelination at this stage. For regulatory CD4 T cells (FoxP3), no significant differences were observed. Although IL-17 measurements in CD4 T cells in **Figure 4d** were not significant, we can include this data in the revised manuscript for completeness. Fluoromyelin staining was performed for symptomatic EAE treatment and demonstrated the myelin-protective effects of intranasal TGF α treatment. Additionally, we have performed NeuN staining for the knockout of microglial TGF α and included these results in the revised manuscript (**Fig. 2I and Fig. 4o**). By including these*

additional data points and analyses, we aim to provide a more comprehensive understanding of TGF α 's protective effects during EAE.

9) For MRI data in Fig. 4C, use of contrast enhancement to measure lesion volume should be qualified and noted to reflect the volume of active inflammation at this recovery time point, not necessarily the total lesion volume (line 207-209).

We thank the reviewer for this assessment. Indeed, our MRI data do represent active lesions as measured by contrast enhancement. To clarify, we have included this information in the methods section to clarify this point (Lines 822-823) and have adjusted the labeling in Fig. 4c.

10) The MS CSF data should specify when samples were taken (during relapse, acute recovery or between attacks) as this has implications for relative TGF α levels contributing to the severity of relapse vs. ongoing recovery/repair.

We have included this information as requested. Indeed, MS patients' samples were taken during relapses, as outlined before. We have discussed potential implications of the timepoint of CSF sampling in the revised version of the discussion (Lines 376-384).

11) MS subjects do not seem ideally age-matched as the average ages of both MS cohorts is younger than controls. Details on how the data was or could be age- and gender-matched would be helpful.

We have now included a list of patients (Table 1a: ELISA, Table 1b: Multiplex analyses) with detailed information on diagnosis, sex, age, disease duration, EDSS, and immunomodulatory therapy in the revised manuscript. Indeed, there is an age difference between the control group (mean age 40.8 years) and RRMS patients (mean age 33.1 years) of 7 years. To address this point, we have included correlations of TGF α levels with age, where we did not detect age-dependency of TGF α levels in the CSF (Fig. Suppl. 4). Still, to address this potential confounder, we have noted this limitation in the revised version of the discussion (Lines 376-384, 388-392).

12) The Discussion would benefit from a more detailed discussion of downstream effectors of TGF α -EGF signaling and how this pathway might have parallel effects 1) steering proinflammatory CD4+ T cell differentiation to IL-17-, IFN γ - and GM-CSF-producing subtypes and 2) driving oligodendrocyte differentiation and repair.

We thank the reviewer for this opportunity. In the revised manuscript, we have expanded the Discussion to include a detailed exploration of the downstream effectors of TGF α -EGF signaling. We elaborated on how TGF α , through EGFR activation, may contribute to both proinflammatory responses and tissue repair mechanisms. We discussed the dual role of EGFR signaling in promoting changes in CD4+ T cell subsets (Th1, Th17, GM-CSF-producing cells) and enhancing oligodendrocyte differentiation and remyelination. This information has been added in the discussion section of the revised version of our manuscript (Lines 328-331).

13) In the Discussion, the limited "availability of..treatments...designed to promote resolution" seems to represent the strongest translational application of this work and can be highlighted by the fact that steroids remain the only clinical option for this, citing data that steroids speed up the time course of clinical recovery during MS relapse but do not change the ultimate disability outcome.

We thank the reviewer for this comment and we have emphasized in the discussion section the limited availability of treatments specifically designed to promote resolution in MS. In the revised manuscript,

we discussed the current limitations of steroid treatments, particularly their inability to prevent long-term disability in MS patients and highlighted the potential of TGF α as a therapeutic agent (Lines 354-357).^{2,3}

14) The statement in the Discussion that CNS penetrating TGF α “preserves neurons and oligodendrocytes and reduces lesion volume in the spinal cord, consequently improving clinical outcome” holds for the acute course of EAE peak and recovery (day 25 post-immunization) in this study, but does not exactly demonstrate its benefit in “chronic neuroinflammation,” as noted in line 74. This statement would be more accurate and carry stronger translational promise if EAE differences with TGF α i.n. were shown to persist to the extremely late recovery time-point. A comparison between high dose steroids and TGF α might even reveal useful pre-clinical information regarding the relative magnitude and duration of these respective treatments.

We appreciate the reviewer’s comments and agree that demonstrating the long-term benefits of TGF α in chronic neuroinflammation is essential. In response, we have conducted additional experiments following knock-out animals till the very late stage of EAE (Suppl. Fig. 2g) and have also analyses focused on myelination in Fig. 4n. However, the effects of steroids on CNS autoimmunity have been documented in animal and clinical studies^{4,5} already, and we believe that these comparational studies, while of potential importance, lie beyond the scope and main message of this manuscript. To still address this point, we have revised the discussion to reflect both the current limitations and the new evidence supporting myelination preservation during late stages of neuroinflammation (Lines 366-394), as well as the relevance of steroids in the resolution of acute deficits in MS.

Reviewer #3

The authors have provided evidence that microglia are the primary source of TGF α in the murine CNS, its production is increased on acute inflammation and reduces with inflammatory resolution, and that TGF α is protective in EAE. Microglia with TGF α -KO in EAE impaired recovery. They detected an increase of infiltrating cells (CD4+ T-cells (with elevated IL-17 and INF γ), macrophages, pro-inflammatory monocytes, pro-inflammatory B cells and neutrophils in TGF α - mice. They also measured a decrease of oligodendrocyte lineage cells including olig2+ oligodendrocytes. They provide further evidence for the protective properties of TGF α by utilising lysolecithin in vitro and in vivo models of demyelination, showing that addition of TGF α reduced demyelination in comparison to vehicle treated controls. Their results suggest that microglial TGF α has a role in recovery after inflammatory damage in EAE, possibly through pathways which regulate pro-inflammatory T cells/myeloid cells. Additionally, they provide evidence for a neuroprotective role for TGF α as oligodendrocyte lineage cell and neuronal cell death was lessened with TGF α addition, consequently, affecting de/remyelination.

They also demonstrate that TGF α can be administered intranasally and that via this delivery method it can penetrate the CNS. Its addition promotes glial-mediated pathways which contribute to neuron survival and clinical recovery in EAE. Importantly, this work may be translatable to human MS as they demonstrate that TGF α is reduced in the CSF of pwRRMS.

The results, methodology and conclusion provided are clear, but a few points need addressing:

We thank the reviewer for his positive assessment of the manuscript and have incorporated additional data to address the concerns brought forward by the reviewer.

1. General – is microglia being used as a catch-all term for microglia and CNS macrophage-like cells? Needs clarification when term first used if so.

*We thank the reviewer for pointing up this important point. In our study, we identify microglia by CD45^{int}CD11b⁺ expression, together with the absence of Ly6C and Ly6G. In contrast, CNS macrophages were identified by CD45^{high}CD11b⁺ expression, positive MHCII expression, Ly6C⁺ expression, and further characterized by the absence of CD11c. We have updated the Methods section to include these specific gating strategies, ensuring precise identification and differentiation of these cell populations, and have pointed out this strategy as well as potential limitations in the revised version of the manuscript (**Lines 712-715, Suppl. Fig. 5**). Should the reviewer feel the need for further adaptation of the nomenclature, we are glad to do so if suggested.*

2. There is a typo in the sentence over lines 492 – 494, this typo makes this portion of the methodology unclear.

*We thank the reviewer for this comment. We have corrected the typo (highlighted in **Line 593**) to clarify the method and apologize for this oversight.*

3. Figure 1 – include which regions of brain/ level of spinal cord were used for analysis of TGF α expression and how many sections were used for the immunohistochemical analysis.

*We thank the reviewer for this point and have included this information in the revised version of the manuscript (**Lines 793-799**).*

4. Include abbreviations & meanings in figure legends, e.g., figure 3 – define RBPMS.

We have added this information to the figure legends and apologize for this oversight.

5. Discussion – lines 307 – 309 “This highlights the pleiotropic effects of TGF α and provides insights into its potential to mitigate disease progression”. This statement is overreaching based on the evidence presented in the paper. The authors include samples from pwRRMS and inflammatory/toxin-induced demyelination models which do not accurately capture/model the progressive phase of human MS. There is evidence in the literature to show that current DMTs, that reduce annualised relapse rate, do not prevent disease progression. Therefore, this conclusion needs to be reworded to reflect that this is not something that can be assumed and instead provides evidence to warrant further investigation in the context of progressive disease.

We thank the reviewer for his comment. We have revised and tuned down this statement in the discussion section of our manuscript (Lines 351-352), and further highlighted the need for additional translational research on the effects of DMTs on disease progression⁶.

This work lacks some translatability as the authors have not provided evidence of TGF α expression in human MS or control CNS tissue. Although this limitation should be addressed, it is outside of the scope of the present study and its exclusion is not sufficient to prevent publication. It is my view, that with the minor alterations listed above, that this work is suitable for publication.

We thank the reviewer for bringing up this point as well as his/her assessment of the publicability of our study. Even though we do agree that this analysis falls outside the scope of our current study, we now refer to our previous studies on TGFA expression in human MS tissue⁷ (Rothhammer et. al., 2018) and have incorporated this information into the revised version of the manuscript (Lines 374-376). We have included this information in the revised version of the discussion.

1. Lim, D., et al. Anti-tumor activity of an immunotoxin (TGF α -PE38) delivered by attenuated *Salmonella typhimurium*. *Oncotarget* **8**(2017).
2. Ciccone, A., et al. Corticosteroids for the long-term treatment in multiple sclerosis. *Cochrane Database Syst Rev*, Cd006264 (2008).
3. Dutt, M., Tabuena, P., Ventura, E., Rostami, A. & Shindler, K.S. Timing of Corticosteroid Therapy Is Critical to Prevent Retinal Ganglion Cell Loss in Experimental Optic Neuritis. *Investigative Ophthalmology & Visual Science* **51**, 1439-1445 (2010).
4. Brusaferri, F. & Candelise, L. Steroids for multiple sclerosis and optic neuritis: a meta-analysis of randomized controlled clinical trials. *J Neurol* **247**, 435-442 (2000).
5. Dutt, M., Tabuena, P., Ventura, E., Rostami, A. & Shindler, K.S. Timing of corticosteroid therapy is critical to prevent retinal ganglion cell loss in experimental optic neuritis. *Invest Ophthalmol Vis Sci* **51**, 1439-1445 (2010).
6. Brown, J.W.L., et al. Association of Initial Disease-Modifying Therapy With Later Conversion to Secondary Progressive Multiple Sclerosis. *JAMA* **321**, 175-187 (2019).
7. Rothhammer, V., et al. Microglial control of astrocytes in response to microbial metabolites. *Nature* **557**, 724-728 (2018).

Rebuttal letter

Reviewer #1:

The manuscript is greatly improved and with all the additional data and discussion I am satisfied that the current version is worthy of publication.

We thank the reviewer for their positive feedback and are grateful that the additional experiments and revisions have addressed the previous concerns. We appreciate the time and thoughtful consideration given to our manuscript.

Reviewer #2:

In referencing data in 4Q, I would recommend changing 'disease severity' to 'disability at the time of relapse'. 'Disease severity' may cause confusion for clinician readers.

We thank the reviewer for this suggestion. We have revised the text accordingly and now refer to this as “disability at the time of relapse” in the revised version of the manuscript to enhance clarity and clinical relevance.

Consider including that a limitation of the study is that the effects of TGF α on CD8⁺ T cells, specifically regulatory CD122⁺ cells, were not explored...

We agree with the reviewer that regulatory CD122⁺ CD8⁺ T cells have been implicated in EAE remission and may represent a relevant effector pathway. While this population was not specifically assessed in our study, we now acknowledge this as a limitation in the Discussion section and propose it as an important avenue for future investigation.

Reviewer #3:

I don't have further comments.

We thank the reviewer for their time and positive evaluation of our revised manuscript.

TGF α controls checkpoints in CNS resident and infiltrating immune cells to promote resolution of inflammation

Supplementary Figure 1. Spatiotemporal regulation of microglia-derived TGF α during acute CNS inflammation. **a**, Immunostaining and quantification (**b**) of TGF α + Iba1+ and TGF α + GFAP+ cells and DAPI for nuclear staining in the optic nerves of naïve and EAE mice. n = 24 per group. Scale bar 50 μ m. **c**, Immunostaining and quantification (**d**) of TGF α + NeuN+ cells and DAPI for nuclear staining in spinal cords of EAE (onset, peak, recovery, late recovery; n=3/5 per timepoint) and naïve mice (n=5). Scale bar 50 μ m. **e**, Relative expression (% of parent) of TGF α + microglia and astrocytes (**f**) in spinal cords and brains of EAE during onset, peak, recovery, late recovery, extremely late recovery and naïve mice (n=3 per timepoint) quantified by intracellular flow cytometry. **g**, UMAP plot of TGF α expression in subsampled microglia in the brain (left) and spinal cord (SC; right) at peak of EAE analysed by high-dimensional flow cytometry. **h**, RT-qPCR analysis of *Tgfa* expression in primary mouse microglia and human microglia (HMC3) (**i**) stimulated with LPS over timecourse (1h, 2h, 4h, 6h, 8h, 24h). n = 3 per group. **j**, RT-qPCR analysis of *Tgfa* expression in primary mouse astrocytes stimulated with TNF α / IL-1 β or vehicle over timecourse (1h, 2h, 4h, 6h, 8h, 24h). n = 4/6 per group. **k**, RT-qPCR analysis of *Tgfa* expression in neuronal cells (N2a) stimulated with TNF α . n = 3/4 per group. **l**, RT-qPCR analysis of *Tgfa* expression in primary mouse oligodendrocytes stimulated with TNF α . n = 3 per group. **m**, UMAP plot of subsampled microglia in the spinal cord (blue) and brain (yellow) during the course of EAE (n=5-6 per timepoint) analysed by high-dimensional flow cytometry. Two-way ANOVA with Sidak's multiple comparisons test in (**b**). Two-way ANOVA with Tukey's multiple comparisons test in (**e**, **f**). One-way ANOVA with Dunnett's multiple comparisons test in (**h**, **l**, **j**). Unpaired t-test in (**k**, **l**). * P < 0.05; ** P < 0.01; *** P < 0.001; **** P < 0.0001.

Supplementary Figure 2. Microglial knock-out of TGF α during acute CNS inflammation. **a**, RT-qPCR analysis of *Tgfa* expression in sorted mouse microglia from the CNS of *CD11b::scrambl* (n=11) and *CD11b::TGF α* (n=5) mice. **b**, Quantification (left) and histogram (right) of mean fluorescence intensity (MFI) of microglial (MG) TGF α expression in the CNS of *CD11b::scrambl* and *CD11b::TGF α* mice analysed at day 30 by intracellular flow cytometry. n = 4 per group. **c**, Immunostaining and quantification (**d**) of TGF α + Iba1+ cells and DAPI for nuclear staining in the spinal cord of *CD11b::scrambl* (n=14) and *CD11b::TGF α* (n=15) mice. Scale bar 30 μ m. **e**, EAE development in mice transduced with *CD11b::scrambl* (n=5) or *CD11b::TGF α* (n=5) plotted from onset of disease. **f**, Delivery of lentiviruses via intracerebroventricular (i.c.v.) injection at peak of EAE and disease development in mice transduced with *CD11b::scrambl* or *CD11b::TGF α* (n=5). **g**, Representative overview images of immunostaining of Olig2+ oligodendrocytes (top), SMI32 (middle) and NeuN+ neurons (bottom, DAPI as nuclear staining) in lumbar spinal cord of *CD11b::scrambl* (n=5) and *CD11b::TGF α* (n=4) mice. **h**, EAE development until day 45 in mice transduced with *CD11b::scrambl* (n=5) or *CD11b::TGF α* (n=4). **i**, Quantification of mean fluorescence intensity (MFI) of microglial (MG) TGF α expression in the CNS of *CD11b::scrambl* and *CD11b::TGF α* mice analysed at day 45 by intracellular flow cytometry. n = 4 per group. **j**, Abundance of monocytes (mono), neutrophils (neutro), pro-inflammatory monocytes (mono), dendritic cells (DCs) and CD4+ T cells, and effector T cells (T eff), regulatory T cells (T reg), and B cells (**k**) in the CNS of *CD11b::scrambl* (n=5) and *CD11b::TGF α* (n=4) mice (day 45). **l**, iNOS production by monocytes (mono) in the CNS of *CD11b::scrambl* (n=5) and *CD11b::TGF α* (n=4) mice analysed by intracellular flow cytometry (day 45). Data shown as mean \pm SD. Data shown as mean \pm SEM in (**e**, **f**, **h**). Non-parametric Wilcoxin matched-pairs signed-rank test in (**e**, **f**, **h**). Unpaired t-test in (**a**, **b**, **d**, **i**, **j**, **k**, **l**). * P < 0.05; ** P < 0.01; *** P < 0.001; **** P < 0.0001.

Supplementary Figure 3. TGF α promotes protective effects under inflammatory and demyelinating conditions. **a**, Purity of primary glial cells after shake off (astrocytes, left), sorted microglia (middle) and oligodendrocytes (right). **b**, RT-qPCR analysis of *Tnf*, *Ccl2*, *Csf2* and *Lif* expression in primary mouse microglia stimulated with vehicle, LPS \pm TGF α or pre-stimulation with LPS + LPS/ TGF α . n = 4 per group. **c**, RT-qPCR analysis of *Nos2* and *Lif* expression in primary mouse astrocytes stimulated with vehicle, TNF α / IL-1 β \pm TGF α or pre-stimulation with TNF α / IL-1 β + TNF α / IL-1 β / TGF α . n = 4 per group. **d**, Schematic, RT-qPCR analysis of *Mbp* expression (**e**) and flow cytometric analysis (**f**) of primary mouse oligodendrocytes during differentiation \pm TGF α , Triiodothyronine (T3), PDGF/FGF. n = 3 per group. **f**, Live (left) and flow cytometric quantification (right) of A2B5+, PDGFR α +, MBP+, and O4+ cells (% of singlets) during differentiation \pm TGF α , Triiodothyronine (T3), PDGF/FGF. n = 3 per group. One-way ANOVA with Tukey's multiple comparisons test in (**b**, **c**, **e**, **f**). * P < 0.05; ** P < 0.01; *** P < 0.001; **** P < 0.0001.

Supplementary Figure 4. TGF α as treatment target for lesion resolution in autoimmune CNS inflammation. **a**, Quantification of mean fluorescence intensity of pEGFR expression in oligodendrocytes (oligo) and endothelial cells (endo) of the CNS from TGF α or vehicle treated EAE mice (intranasal) analysed by intracellular flow cytometry. n = 4 per group. **b**, CNS cells from TGF α or vehicle treated EAE mice (intranasal) analysed by high-dimensional flow cytometry showing relative expression levels (% of parent) of EGFR in astrocytes. n = 5 per group. **c**, Enzyme-linked Immunosorbent Assay (ELISA) of whole brain samples from TGF α or vehicle treated EAE mice (intranasal). **d-e**, Abundance of CD4+, CD8+ T cells (**d**) and dendritic cells (DC), monocytes (mono) and pro-inflammatory monocytes (MHCII+) (**e**) in spleen from TGF α or vehicle treated EAE mice (intranasal) analysed by high-dimensional flow cytometry. n = 4 per group. **f**, Relative expression (% of parent) of iNOS in Ly6C+ splenic monocytes from TGF α or vehicle treated EAE mice (intranasal) analysed by intracellular flow cytometry. n = 4 per group. **g**, Splenic cells from TGF α or vehicle treated EAE mice (intranasal) analysed by intracellular flow cytometry showing expression levels of proliferation (Ki67) marker, transcription factor (FoxP3) and cytokine production (IFN γ , IL17a, GM-CSF) in CD8+ and CD4+ (**h**) T cells. n = 4 per group. **i**, Quantification of EGFR+ GLAST+ cells in the lumbar spinal cord of *GFAP::scrambl* (n=15) and *GFAP::ErbB1* (n=15) mice analysed by immunostaining. **j**, CNS cells from TGF α or vehicle treated EAE mice (intranasal; start day 7) transduced with *GFAP::ErbB1* (n=4) analysed by intracellular flow cytometry showing IL17+ production in monocytes and CD24+ production in CD4+ T cells. **k**, Representative overview images of immunostaining of Olig2+ oligodendrocytes (top), SMI32 (middle) and NeuN+ neurons (bottom, DAPI as nuclear staining) in lumbar spinal cord of symptomatic intranasal treatment with TGF α or vehicle. **l**, Solid line shows linear regression of correlation between TGFA concentration (pg/ml) measured by Enzyme-linked Immunosorbent Assay (ELISA) and age in control samples (n=17). Values are means of technical duplicate measurements Unpaired t-test in (**a**, **b**, **c**, **i**). Two-way ANOVA with Sidak's multiple comparisons test in (**d**, **e**, **f**, **g**, **h**, **j**). * P < 0.05; ** P < 0.01; *** P < 0.001; **** P < 0.0001

Supplementary Figure 5. Gating strategy: Flow cytometry of CNS surface marker.

Supplementary Table 1: Characteristics of individual MS patients and controls.

a, Characteristics of MS patients and controls used in ELISA measurement of TGFA. **b**, Characteristics of MS patients used for multiplex analysis. No additional relevant comorbidities or pharmaceutical treatments were reported in either group.